# Magnetothermal nanoparticle technology alleviates parkinsonian-like symptoms in mice

Sarah-Anna Hescham[1✉], Po-Han Chiang[2,3], Danijela Gregurec [2,4], Junsang Moon[2,5], Michael G. Christiansen[6], Ali Jahanshahi[1], Huajie Liu [1], Dekel Rosenfeld[2], Arnd Pralle [7], Polina Anikeeva [2,5,8,9] & Yasin Temel[1,9]

Deep brain stimulation (DBS) has long been used to alleviate symptoms in patients suffering from psychiatric and neurological disorders through stereotactically implanted electrodes that deliver current to subcortical structures via wired pacemakers. The application of DBS to modulate neural circuits is, however, hampered by its mechanical invasiveness and the use of chronically implanted leads, which poses a risk for hardware failure, hemorrhage, and infection. Here, we demonstrate that a wireless magnetothermal approach to DBS (mDBS) can provide similar therapeutic benefits in two mouse models of Parkinson's disease, the bilateral 1-methyl-4-phenyl-1,2,3,6-tetrahydropyridine (MPTP) and in the unilateral 6-hydroxydopamine (6-OHDA) model. We show magnetothermal neuromodulation in untethered moving mice through the activation of the heat-sensitive capsaicin receptor (transient receptor potential cation channel subfamily V member 1, TRPV1) by synthetic magnetic nanoparticles. When exposed to an alternating magnetic field, the nanoparticles dissipate heat, which triggers reversible firing of TRPV1-expressing neurons. We found that mDBS in the subthalamic nucleus (STN) enables remote modulation of motor behavior in healthy mice. Moreover, mDBS of the STN reversed the motor deficits in a mild and severe parkinsonian model. Consequently, this approach is able to activate deep-brain circuits without the need for permanently implanted hardware and connectors.

[1] Department of Neurosurgery, Mental Health and Neuroscience, Maastricht University Medical Center, Maastricht, The Netherlands. [2] Research Laboratory of Electronics and McGovern Institute for Brain Research, Massachusetts Institute of Technology, Cambridge, MA, USA. [3] Institute of Biomedical Engineering, National Yang Ming Chiao Tung University, Hsinchu, Taiwan, ROC. [4] Department of Chemistry and Pharmacy, Chair of Aroma and Smell Research, Friedrich-Alexander-Universität Erlangen-Nürnberg, Erlangen, Germany. [5] Department of Materials Science and Engineering, Massachusetts Institute of Technology, Cambridge, MA, USA. [6] Department of Health Sciences and Technology, ETH Zürich, Zürich, Switzerland. [7] Department of Physics, University at Buffalo, Buffalo, NY, USA. [8] Department of Brain and Cognitive Sciences, Massachusetts Institute of Technology, Cambridge, MA, USA. [9]These authors contributed equally: Polina Anikeeva, Yasin Temel. ✉email: sarah.hescham@maastrichtuniversity.nl

Deep-brain stimulation (DBS) is an invasive treatment involving the implantation of electrodes, which deliver electrical impulses to specific parts of the brain. It has substantial therapeutic effects in a range of disorders, including Parkinson's disease (PD), essential tremor, and dystonia[1–4]. In particular, high-frequency DBS of the subthalamic nucleus (STN) is now considered a treatment of choice for patients suffering from severe forms of PD with motor fluctuations and major side effects of dopaminomimetic treatments[5]. Nevertheless, the fundamentals of DBS hardware and software design have seen limited progress since its advent three decades ago and the main drawback of current DBS is that it requires a wired and chronically implanted system. As a result, many patients are reluctant to undergo DBS even when treatment is warranted[6]. Recent advances in the field of neuromodulation stress the urgency of developing wireless deep-brain stimulation treatments[7]. Existing non-invasive brain stimulation methods include repetitive transcranial magnetic stimulation (rTMS), transcranial direct current stimulation (tDCS), and focused ultrasound among others. In rTMS, high-intensity magnetic fields are required to stimulate deep-brain regions, and these may lead to facial pain, facial and cervical muscle contractions, and other undesirable side effects[8]. Furthermore, the scattering and absorption of electromagnetic waves and ultrasound within brain tissue limits the spatial resolution and the penetration depth of these fields[9]. These side effects are likely caused by off-target neurostimulation. Neither DBS nor these non-invasive techniques provide any cellular targeting. To address this, we explored the therapeutic potential of a remote magnetic nanoparticle neuromodulation technique which does not require any tether into the brain and yet can provide cell-specific targeting. Magnetothermal DBS (mDBS) activates heat-sensitive ion channels in the brain, such as the transient receptor potential cation channel subfamily V member 1 (TRPV1)[10,11] using magnetic nanoparticles (MNPs), which dissipate heat via hysteretic power losses in an alternating magnetic field (AMF; Fig. 1A)[12]. Magnetothermal neuromodulation was first demonstrated in vitro and in C. elegans a decade ago[13], and more recently applied in mice. Anaesthetized animals were used at first[14], and later involuntary behaviors were evoked in freely moving mice[15].

Here we sought to validate mDBS as a wireless therapeutic in parkinsonian mice. In particular, we wanted to demonstrate remote neuromodulation using MNPs that are optimized for efficient heat dissipation at clinically relevant AMF conditions (i) in vitro, (ii) in freely moving, wild-type mice, and (iii) in a mild and severe parkinsonian mouse model. Although our study relied on exogenously expressed TRPV1, this channel is endogenously expressed in neurons and glia cells in certain regions of the mammalian central nervous system[16]. Furthermore, MNPs with chemistries similar to those of clinically approved contrast agents for magnetic resonance imaging (MRI) exhibit minimal cytotoxicity and remain intact several months after injection[14,17,18]. In fact, the long-term effectiveness of magnetothermal stimulation has been recently demonstrated after two months postinjection, suggesting a non-significant decrease of MNPs concentration at the injection site and a non-significant change to their magnetic properties[19]. Consequently, this wireless minimally invasive neuromodulation technology may pave the way for future human applications with greater patient adherence to DBS treatment recommendations[20–22].

## Results

### Specific loss of power of iron-oxide nanoparticles

We first identified the most optimal AMF settings to induce heat dissipation in MNPs. For this, we measured the specific loss power (SLP) of these MNPs using a custom AMF coil driven by a resonant circuit[23,24]. The measurements were conducted under an AMF with frequency $f = 160$ kHz and amplitudes $H_0 = 10, 20, 30, 40,$ and $50$ kA/m. The SLP initially increases with the increasing AMF amplitude and then plateaus at $558 \pm 25$ W/g$_{[Fe]}$ at amplitudes approaching the coercive field of the particles (Fig. 1B). For all subsequent in vitro and in vivo experiments, we thus applied a field with $H_0 \geq 30$ kA/m.

### In vitro magnetothermal control

In order to demonstrate magnetothermal control in vitro, we evaluated intracellular Ca$^{2+}$ influx in human embryonic kidney (HEK293FT) cells upon AMF stimulation. HEK293FT cells were first transfected with the TRPV1 transgene. For this, TRPV1 was placed under the excitatory neuronal promoter calmodulin kinase II α-subunit along with mCherry, which was separated from TRPV1 by the post-transcriptional cleavage linker p2A (CaMKIIα::TRPV1-p2A-mCherry) and packed into a lentiviral vector. Three days later, cells were additionally transfected with the adeno-associated virus serotype 9 (AAV9) carrying GCaMP6s under the neuronal promoter human synapsin (hSyn::GCaMP6s) for measurement of intracellular Ca$^{2+}$ changes. In vitro, magnetothermal stimulation was performed 5 days later. We used a custom-built coil, consisting of a toroidal soft ferromagnetic core with a 7.5 mm gap and wrapped with litz wire, to apply an AMF stimulus[24]. The field magnitude was measured by an inductive pickup loop of known geometry and an oscilloscope. Since we have done numerous in vitro experiments using MNPs <25 nm that show the greatest SLP when the AMF is set to 522 kHz and 15 kA/m in the past (Chen et al.[14]), here, we merely wanted to verify that changing the frequency to 160 kHz in order to maximize the SLP for our MNPs would replicate our previous results. For more information on AMF condition selection please see ref. [25]. For this reason, we conducted a single experiment, out of which 30 cells were randomly selected. HEK293FT cells were shortly incubated in either Tyrode (control) or 2 mg/ml ferrofluid in Tyrode and stimulated for 20 s with AMF at 160 kHz and 30 kA/m during 60 s long fluorescence recordings (no field 0–20 s, AMF 20–40 s, no field after 40 s). Only cells incubated in MNP solution responded to the field stimulus ($f = 160$ kHz, $H_0 = 30$ kA/m), whereas cells incubated in Tyrode did not exhibit changes in intracellular Ca$^{2+}$ concentrations (repeated-measures ANOVA of the different time points: $F(180,10440) = 19.53$, $p < 0.0001$, Fig. 1C). A field-induced temperature increase in the MNPs solution triggered a GCaMP6s fluorescence intensification ($\Delta F/F_0$), which differed significantly between 25 and 41 s between groups as revealed by a Bonferroni post hoc test ($p \leq 0.01$). The other time points, during which the AMF was off, were not significant (Fig. 1C). Since we do not have repeated experiments, we also performed a Tukey analysis and show the significant difference in response between cells with MNPs and without MNPs at $p < 0.001$ (Fig. 1D). This is in line with our previous study, in which an AMF of a different frequency and amplitude was used[14].

### In vivo magnetothermal neuromodulation of mouse behavior

To further evaluate the biomedical potential of mDBS, we applied it to stimulate the STN due to its clinical relevance as a target for DBS treatment of movement disorders. It has been shown before that classical unilateral STN DBS can induce circling behavior in rodents[26] and therefore we also aimed to assess rotational behavior in response to unilateral STN mDBS. Rotational motor behavior offers an objective read-out and is also compatible with laboratory-scale magnetic field coils. We first heat-sensitized neurons in the STN through lentiviral delivery of TRPV1, which was followed by MNP injection ($n = 7$) into the same region

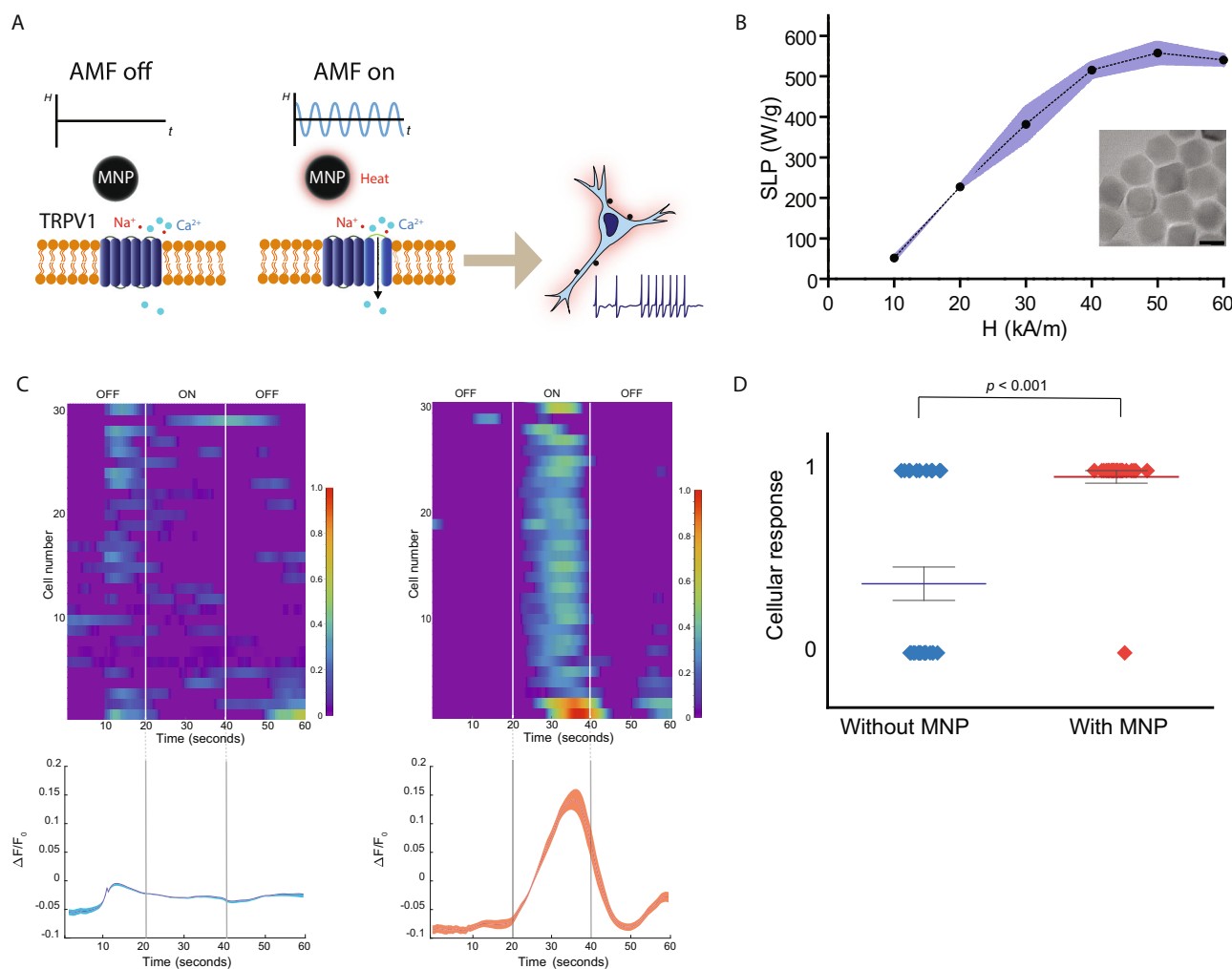

**Fig. 1 In vitro magnetothermal control. A** Experimental scheme. Magnetic field stimulation ("AMF ON") induces MNP heating and thereby membrane depolarization. **B** Specific loss power (SLP, normalized to iron content [Fe]) measurements at 160 kHz and different field amplitudes. The SLP was calculated from the rate of temperature increase of a 1 mg/ml [Fe] sample. Data presented as mean ± S.D., $n = 3$ independent experiments for each field amplitude. Transmission electron micrograph of MNPs, scale bar is 20 nm. **C** Heatmaps of fluorescence intensity changes recorded from GCaMP6s transients observed in TRPV1-expressing HEK293FT cells during magnetic field stimulus with and without MNPs (for each condition $n = 30$ cells). Normalized fluorescence intensity change ($\Delta F/F_0$) as a function of time (dark lines indicate the mean, and shaded areas indicate S.E.M.). Fluorescence increase was observed only in TRPV1+ cells with MNPs upon AMF application. **D** Value "0" represents cells not responding to AMF and value "1" are cells responding to the AMF stimulation. We show the significant difference in response between cells with MNPs and without MNPs as confirmed by one-way ANOVA followed by Tukey post hoc analysis (F(1,56) = 35.805, $p < 0.001$ (*)). Data presented as mean ± S.E.M. Source data are provided as a Source data file.

6–8 weeks later (Fig. 2A). Control mice ($n = 10$) were subjected to the same viral delivery and surgical procedures, but were injected with non-magnetic nanoparticles (non-MNPs) composed of wüstite (FeO)[19]. The mice were allowed to recover for 1 week prior to behavioral testing. The circular arena was made out of plexiglass (Ø 9 mm, height 10 cm, fitting within the magnetic coil) and mice were stimulated continuously for 3 min at $f = 160$ kHz, $H_0 = 30$ kA/m. Rotations were classified as ipsilateral or contralateral rotations around the body axis. The frequency of rotations was assessed per 3 min and per 30 s time bins. Mice injected with MNPs unilaterally in the STN showed significantly more contralateral rotations around the body axis compared to control mice (t(15) = 2.052, $p = 0.029$; Fig. 2B, Movie S1). When the 3 min of mDBS were divided into 30 s time bins, the repeated-measures ANOVA indicated a significant effect between groups for contralateral rotations (F(1,15) = 6.307, $p = 0.024$, Fig. 2C). One-tailed $t$-tests revealed the significant difference between mDBS and control mice with regard to

contralateral rotations around the body axis within the time intervals of 30–60 s (t(15) = 2.703, $p = 0.008$), 90–120 s (t(15) = 2.078, $p = 0.027$), and 180–210 s (t(15) = 2.066, $p = 0.035$) when the coil was already turned off. No significant difference was seen in ipsilateral rotations around the body axis (t(15) = 0.755, n.s., Fig. 2B and C). In mice injected with MNPs ($n = 7$), neural activity was triggered across motor regions (regions involved in control of the motor system can be found in Fig. 2D), manifesting in a significantly higher proportion of cells expressing an immediate early gene c-Fos (marker of recent neural activity) when compared to mice injected with non-MNPs ($n = 6$). In particular, we found a significant effect in the primary motor cortex (t(11) = 3.338; $p = 0.007$) and the red nucleus (t(11) = 3.293; $p = 0.007$, Fig. 2E and F). In non-motor regions such as the amygdala, there was no statistical difference between mDBS and control mice injected with non-MNPs (t(11) = 0.771; n.s, Fig. 2E and F). To consider the spatial extent of expected magnetothermal stimulation around the MNPs, we employed a

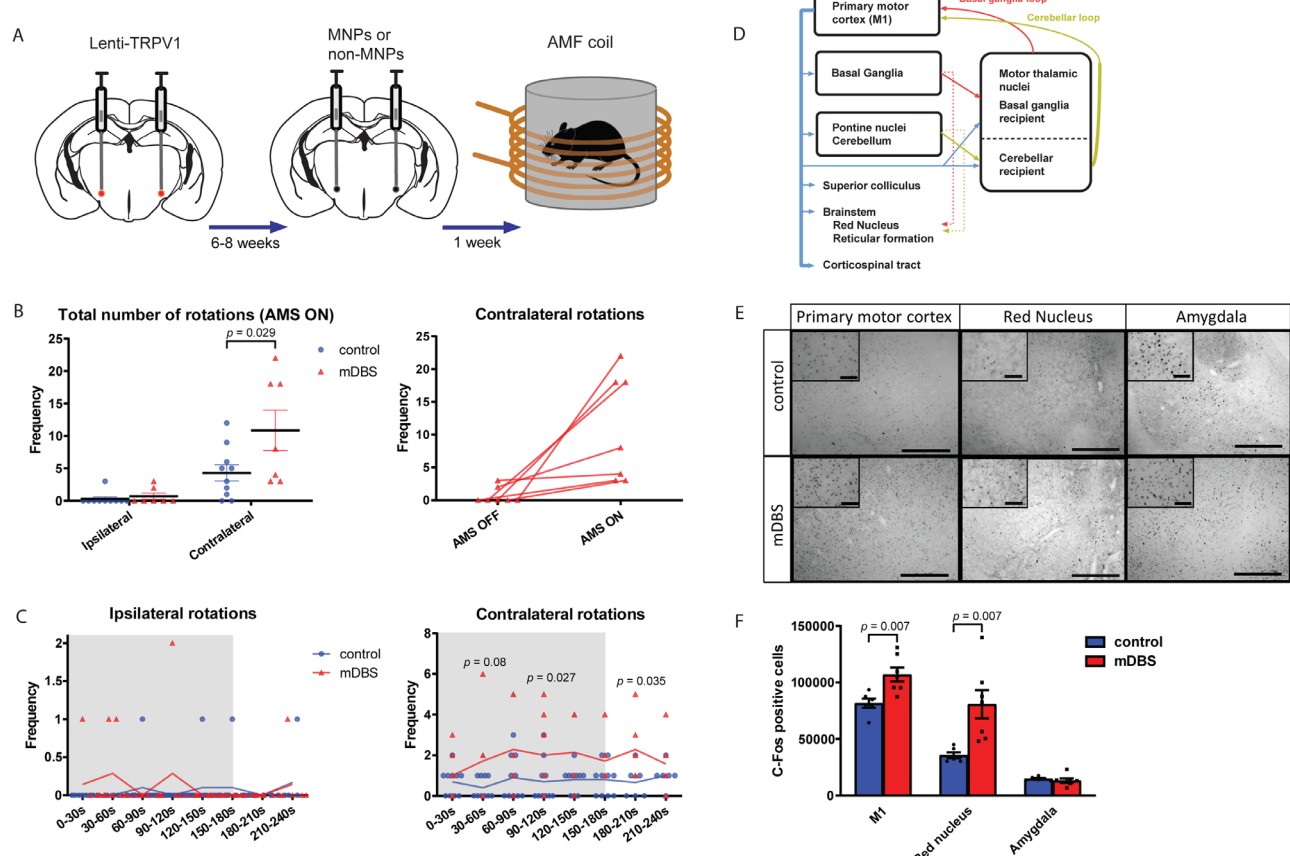

**Fig. 2 Remote magnetothermal neuromodulation of mouse behavior. A** In vivo experimental scheme. **B** Total rotations of STN mDBS ($n = 7$) and control animals ($n = 10$) during AMF ON and OFF conditions. Data presented as mean ± S.E.M., one-tailed $t$-test, t(15) = 2.052, $p = 0.029$. **C** Rotational behavior presented in 30 s time bins. Shaded areas indicate when AMF was on. Repeated-measures ANOVA indicated a significant effect between groups for contralateral rotations (F(1,15) = 6.307, $p = 0.024$, **C**). One-tailed $t$-tests revealed the significant difference between mDBS and control mice with regard to contralateral rotations around the body axis within the time intervals of 30–60 s (t(15) = 2.703, $p = 0.008$), 90–120 s (t(15) = 2.078, $p = 0.027$) and 180–210 s (t(15) = 2.066, $p = 0.035$) when the coil was already turned off. No significant difference was seen in ipsilateral rotations around the body axis (t(15) = 0.755, n.s.). **D** Major central nervous system regions involved in control of the motor system (adapted from ref. [50]). The primary motor cortex (M1) has subcortical descending pathways to the corticospinal tract. Axons of these pathways also target subcortical motor nuclei, such as the red nucleus and brainstem reticular formation, which serve as the origin of the descending rubrospinal and reticulospinal tracts. Axonal collaterals target structures including the pontine nuclei, the superior colliculus, the basal ganglia, and the thalamus. Basal ganglia (red) and cerebellum (gold) form two major loops which can influence M1 and thus motor function. **E** Representative low-power photomicrographs (scale bar = 1000 µm) of coronal brain sections stained for c-Fos (K-25) showing the primary motor cortex M1, red nucleus, and amygdala each with high-power photomicrograph insets in the upper left corner (scale bar = 50 µm) in mice receiving STN mDBS ($n = 7$) and controls ($n = 6$). **F** Comparisons were made between STN mDBS ($n = 7$) and control mice ($n = 6$). STN mDBS animals showed increased Fos expression in motor areas such as M1 and the red nucleus, whereas the amygdala, a non-motor area, was not affected. Data presented as mean ± S.E.M., two-tailed independent samples $t$-test, M1: t(11) = 3.338; $p = 0.007$; red nucleus: t(11) = 3.293; $p = 0.007$; amygdala: t(11) = 0.771; n.s. Source data are provided as a Source data file.

computational model (Supplementary Materials Table S1, Figs. S1, and S2).

**In vivo magnetothermal stimulation can alleviate mild and severe parkinsonian symptoms in freely moving mice.** Finally, we sought to investigate whether the effects of mDBS can be therapeutic. The most common indication for traditional clinical DBS with pronounced beneficial effects is STN stimulation in PD patients with medically refractory levodopa-induced motor complications. Therefore, we applied mDBS to the STN in a 1-methyl-4-phenyl-1,2,3,6-tetrahydropyridine (MPTP) mouse model of PD and in the 6-hydroxydopamine (6-OHDA) hemi-parkinsonian mouse model. For this, we heat-sensitized STN neurons through bilateral (MPTP model) or unilateral (6-OHDA model) lentiviral delivery of TRPV1, which was followed by MNP

or non-MNPs injection into the same region 7 weeks later (Fig. 3A and 4A). A representative schematic illustrating the different areas of transfected cells and the distribution of MNPs can be found in Fig. 3B. For the MPTP model, mice were injected with either saline or MPTP ($2 \times 18$ mg/kg in saline, intraperitoneally), such that there were 4 groups in total: naive controls ($n = 8$), naive mDBS ($n = 10$), MPTP controls ($n = 6$), MPTP mDBS ($n = 7$). Following a 2 weeks recovery, all the mice were placed into the AMF coil and stimulated for 3 min at $f = 160$ kHz, $H_0 = 30$ kA/m. After 3 min, they were taken out of the coil and subsequently tested in the open field for 5 min. The open field test provides a quick assessment of general locomotor activity in a novel and open environment. When considering the total 5 min, there was no significant difference between the groups (Fig. 3B), however, when dividing the total time into 30 s time bins, we noted that MPTP mDBS mice gradually decreased

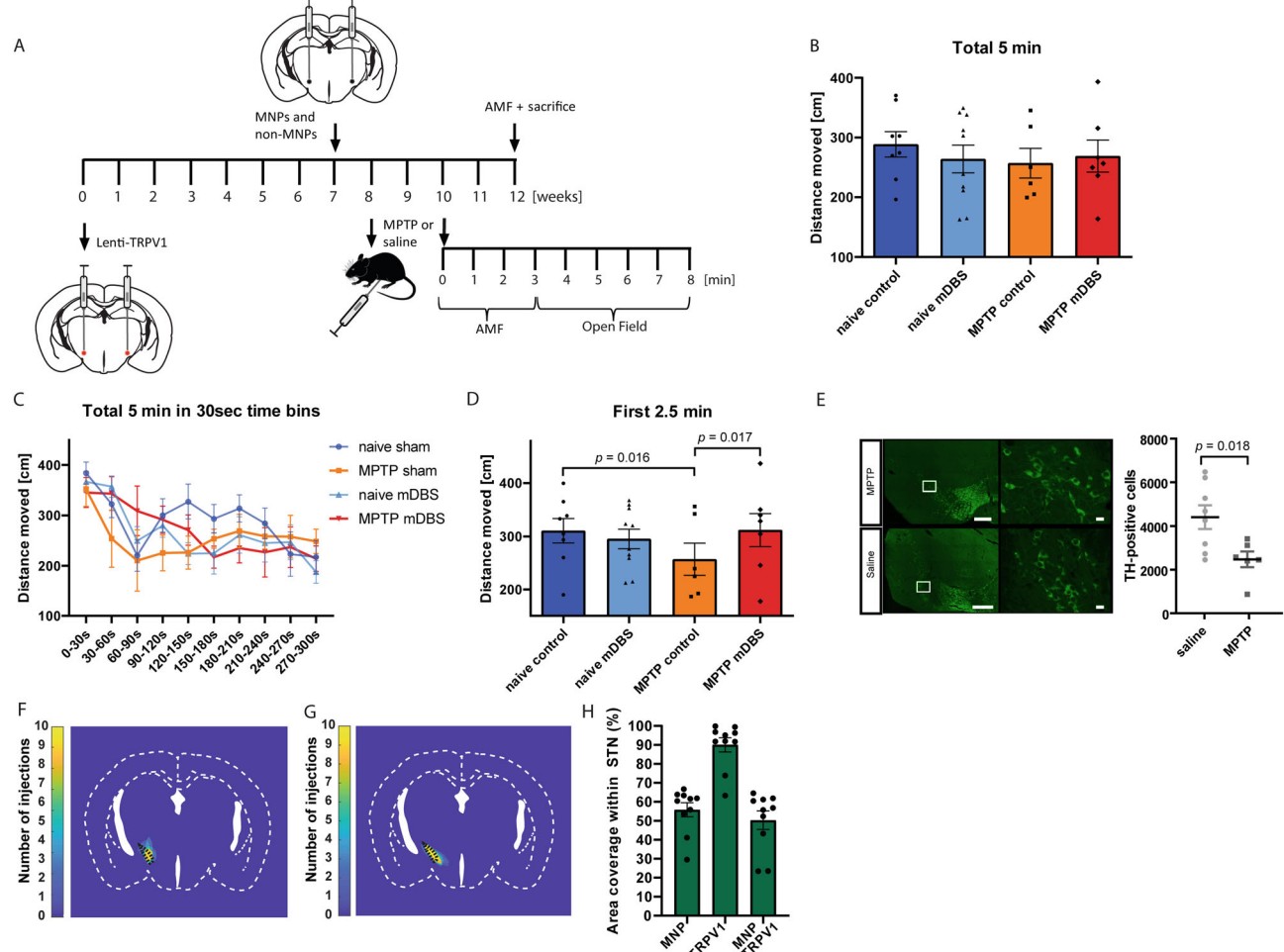

**Fig. 3 Magnetothermal STN DBS alleviates mild parkinsonian symptoms. A** In vivo experimental scheme for the translational study. **B** Total distance moved (in cm) in the open field ± S.E.M. for naive control ($n = 8$), naive mDBS ($n = 10$), MPTP control ($n = 6$), and MPTP mDBS ($n = 7$) mice. There was no significant difference between the groups when considering the total 5 min, one-way ANOVA: F(3,305) = 1.46, n.s. **C** When presenting the data in 30 s time bins, it is apparent that MPTP mDBS animals reduced locomotor behavior after the first 2.5 min of the open field. Data presented as mean ± S.E.M. **D** Total distance moved (in cm) in the first 2.5 min of the open field ± S.E.M. There was a significant difference between naive controls and MPTP controls, as well as MPTP controls and MPTP mDBS mice as confirmed by one-way ANOVA followed by Fisher's least significance test: F(3,150) = 2.488, $p = 0.016$ and $p = 0.017$, respectively. **E** Representative photomicrographs of TH-positive cells for saline ($n = 8$) and MPTP-treated ($n = 6$) animals in the substantia nigra pars compacta. Left: scale bar = 1000 μm, right: scale bar 50 μm. MPTP-treated animals show a 40% reduction in TH levels, when compared to saline-treated mice. Data presented as mean ± S.E.M., two-tailed independent samples $t$-test (t(12) = 2.752, $p = 0.018$). **F, G** Injection maps of **F** MNPs and **G** TRPV1 in the brain ($n = 10$ mice) and demonstrating the coverage areas within the STN (marked in black). **H** Percentage of area covered by MNPs, TRPV1, or both in the STN of $n = 10$ mice. Data presented as mean ± S.E.M. Source data are provided as a Source data file.

their locomotor activity (Fig. 3C). This might be related to cooling of the MNPs injections back to the physiological temperature when the mice are removed from the AMF coil. Therefore, we also considered only the first 2.5 min of the open field, and the Fisher's least significance post hoc test revealed significant motor effects between naive controls and MPTP controls ($p = 0.016$), as well as MPTP controls and MPTP mDBS mice ($p = 0.017$, Fig. 3D). We found that intraperitoneal MPTP injections resulted in reduced bilateral TH-expression in the substantia nigra pars compacta (SNc, two sections averaged per mouse) when compared to controls (t(12) = 2.752, $p = 0.018$, Fig. 3E).

For the 6-OHDA model, in the same stereotactic surgery delivering MNPs to the STN, 6-OHDA was injected in the medial forebrain bundle (Fig. 4A). Due to a high drop-out of animals following 6-OHDA lesion, we have chosen a within-subjects design and tested animals ($n = 10$) before and after 3 min AMF stimulation in the open field, rotarod, and the automated

CatWalk XT gait analysis system. We did not detect any changes in the first 2.5 min of the open field when comparing mice in AMF off and AMF on conditions (Fig. 4C). We did, however, find significant effects of AMF stimulation on the rotarod with regard to motor performance. The rotarod measures the ability of a mouse to maintain itself on a rod that turns at accelerating speed and thus provides a measure of balance, coordination, physical condition, and motor-planning. We found that mice that were stimulated in an AMF were able to stay on the rod for a longer period of time when compared to the AMF off condition (F(1,9) = 8.418; $p = 0.018$; Fig. 4D). We also found significant effects of AMF stimulation for gait- and balance-related general, coordination, static and dynamic parameters during the CatWalk test. These include run duration ($p = 0.036$; Fig. 4E), number of steps ($p = 0.029$; Fig. 4F), step pattern regularity ($p = 0.001$; Fig. 4G), support diagonal ($p = 0.001$; Fig. 4H), support three ($p = 0.007$), stride length ($p = 0.001$, Fig. 4I), print area ($p \le 0.04$; Fig. 4J), max contact area ($p \le 0.012$), print length ($p = 0.001$),

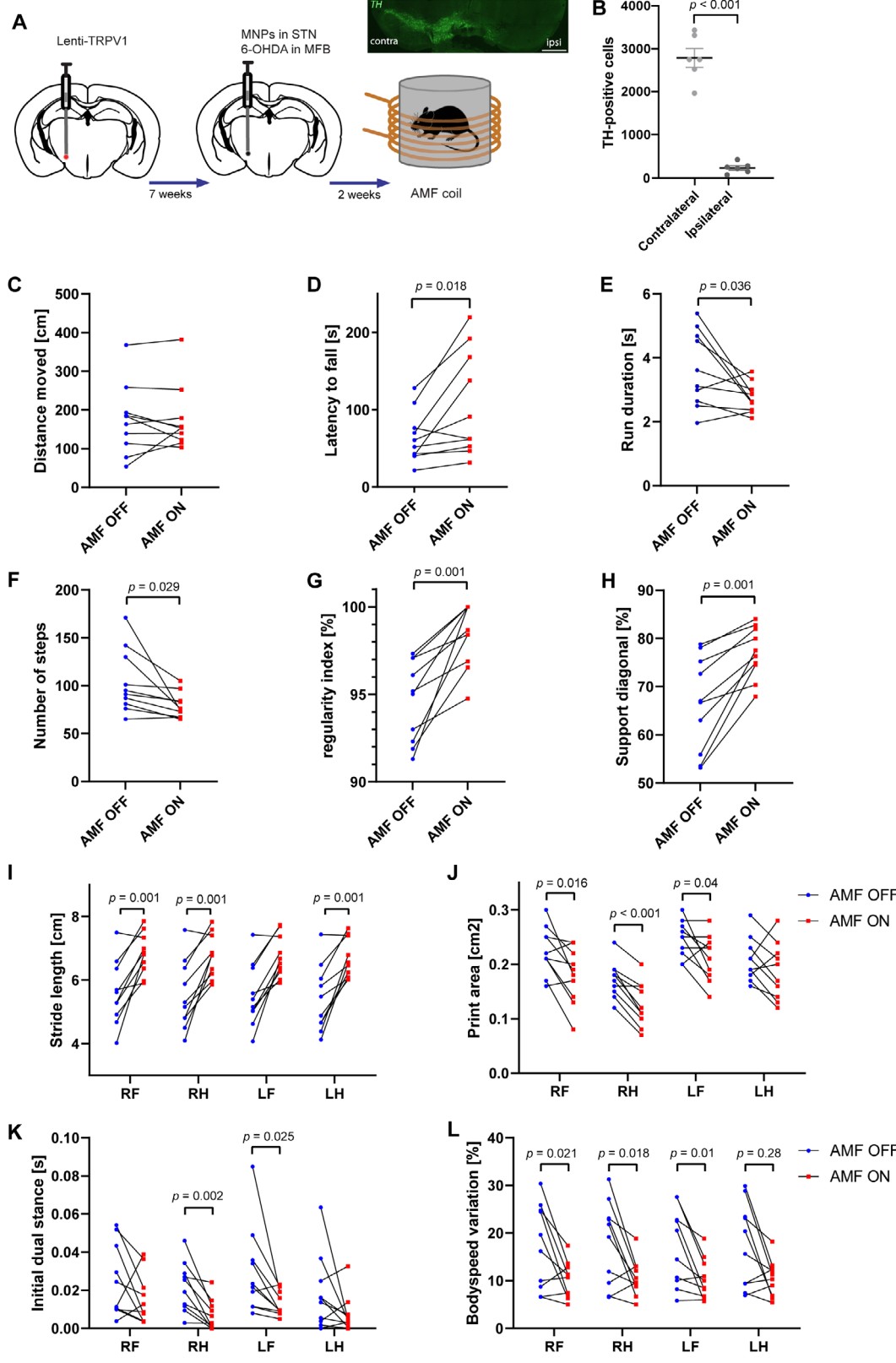

## Discussion

print width ($p = 0.001$), initial dual stance ($p \leq 0.025$, Fig. 4K), terminal dual stance ($p = 0.012$), and body speed variation ($p \leq 0.028$; Fig. 4L). We found that unilateral 6-OHDA injections resulted in 90% reduced TH-expression in the ipsilateral SNc when compared to the contralateral hemisphere (three sections averaged per mouse) ($t(32) = 8.417$, $p < 0.001$, Fig. 4B).

Our work demonstrates that mDBS of the STN effectively activates specific neuronal circuits in parkinsonian mice to alleviate motor disability rapidly and robustly. Developing less-invasive avenues for DBS technology, while increasing the sensitivity and selectivity for DBS target structures has been topic of active

**Fig. 4 Magnetothermal STN DBS alleviates severe parkinsonian symptoms. A** In vivo experimental scheme for the translational study. **B** Representative photomicrographs of TH-positive cells in the substantia nigra pars compacta contralateral and ipsilateral to the 6-OHDA lesion. Scale bar 1000 μm. 6-OHDA lesion resulted in a 90% reduction in TH levels on the ipsilateral substantia nigra pars compacta, when compared to the contralateral side. Data presented as mean ± S.E.M., two-tailed independent samples $t$-test; t(32) = 8.417, $p < 0.001$. **C** Total distance moved (in cm) in the open field ± S.E.M. There was no significant difference between AMF off and AMF on conditions when considering the first 2.5 min. **D** Latency to fall from the rotarod (in s) during AMF off and AMF on conditions averaged from three consecutive trials. Mice that were stimulated in an AMF 3 min before the rotarod, significantly improved their motor performance (repeated-measures ANOVA; F(1,9) = 8.418; $p = 0.018$). **E–L** Representative CatWalk results. AMF stimulation resulted in faster run duration ($p = 0.036$) (**E**), less number of steps ($p = 0.029$) (**F**), increased step sequence regularity (%) ($p = 0.001$) (**G**), increased diagonal limb support ($p = 0.001$) (**H**), increased stride length ($p = 0.001$) (**I**), decreased paw print area ($p \leq 0.04$) (**J**), decreased initial dual stance ($p \leq 0.025$) (**K**), and decreased body speed variation ($p \leq 0.02$) (**L**) as revealed by repeated-measures ANOVA F(1,9) ≥ 5.72. RF: right front paw, RH: right hind paw, LF: left front paw, LH: left hind paw. Source data are provided as a Source data file.

research. Especially in the treatment of advanced PD, DBS of the STN has delivered benefits exceeding those afforded by drug regimens[5]. The STN is a key structure within in cortico-basal ganglia-thalamo-cortical circuits and therefore has a major function in controlling movement[27]. Efferents from the STN to the basal ganglia consist of glutamatergic projections innervating the globus pallidus and the substantia nigra[28].

Here, we targeted the STN with mDBS and assessed its biomedical potential as wireless alternative to classical DBS. Although there is precedent for applying nanomaterials-mediated neuromodulation in clinically-inspired contexts, including photothermal approaches using gold nanoparticles[29,30], there have been no prior reports of translational studies aiming to treat parkinsonian symptoms using mDBS.

Our study relied on exogenously expressed TRPV1 to create uniform levels across mice, which might represent a major challenge for clinical applications. This channel, however, is also endogenously expressed in neurons and glia in certain regions of the mammalian central nervous system[16]. Various studies have shown that TRPV1 can be found in prefrontal cortex, amygdala, hypothalamus, periaqueductal gray, locus coeruleus, cerebellum, hippocampus, and dentate gyrus[31]. Whether TRPV1 is also endogenously expressed in the STN is yet to be elucidated. Notably, TRPV1 receptors can mediate a presynaptic form of long-term depression (LTD) at glutamatergic synapses onto CA1 inhibitory interneurons in the hippocampus[31]. In this respect, endogenous lipid ligands such as anandamide (AEA), can trigger a form of postsynaptic LTD. In the present study, we did not observe any abnormal behaviors in mice transduced to overexpress TRPV1 in the STN neurons. Moreover, control mice followed the same surgical procedure including an injection of a TRPV1-carrying vector. Nevertheless, to rule out possible side effects caused by endogenous AEA, future studies could generate AEA insensitive TRPV1 mutants. Another point of consideration with regard to clinical application is the use of AMF coils. Magnetic coils suitable for cell-destructive therapy in magnetic hyperthermia and magnetothermal neuromodulation in rodents can be engineered to efficiently generate appropriate AMF conditions temporarily over small experimental volumes[15,24]. For chronic use, however, a portable coil on the animal/patient is necessary to maintain therapeutic benefits. In line with this, scaling AMF coils to volumes necessary for neuromodulation in human patients also presents a formidable challenge, as the power requirements to achieve comparable AMF conditions increase substantially[24,32].

To evaluate the extent of the magnetothermal stimulus in the STN and its ability to drive rotational behavior akin to classical DBS, we have previously applied a finite element (FEM) model to calculate the temperature distribution within a ferrofluid infusion in brain tissue using the Penne's bioheat equation[14,19]. In the FEM simulated data, sufficient heating of MNP injection in deep-brain tissue is achieved following a latency of 10 s. More delay

might be caused by a weaker average field in the center of the coil, which together might explain similar performance of mDBS and control animals in the first 30 s time bin of the rotational behavior test (Fig. 1C). Furthermore, the microliter-volume MNP injections cool back to the physiological temperature within 60 s, thereby producing a carry-over effect even when the AMF has been turned off. Additional studies are needed to characterize the precise heating rates per gram of these MNP injections in brain tissue. In a recent study, magnetothermal genetic stimulation of the dorsal striatum has also been reported to induce rotations around the body axis[15]. In this study, the MNPs were bound to the neuronal cell membranes through genetic targeting with antibodies, which may facilitate heat transfer to the heat-sensitive ion channels. Our paradigm, however, does not rely on nanoscale heat transport at the membranes but rather uses bulk heating from a nanoparticle solution in close proximity to the neuronal cells (Fig. S3). It would be interesting to investigate whether binding the MNPs to the cell membrane could further reduce the delay of effective heating in brain tissue and/or reduce the concentration of nanoparticles needed to evoke a behavioral response. These studies, however, motivate further investigation of heat dissipation from individual nanoparticles and solutions to the challenges associated with heat transfer at the nanoscale[33,34].

Next, we applied mDBS of the STN in a mild MPTP mouse model of PD, and found that parkinsonian mice stimulated with mDBS became comparable to controls with regard to general locomotor activity, despite 40% loss of TH-positive cells in the substantia nigra pars compacta (SNc). A general decline in motor frequency or ability to initiate movement comprises bradykinesia, a cardinal symptom of PD. Loss of TH at bradykinesia onset in PD is ~80% in striatum and 40% in the SNc, with similar loss estimated in PD patients[35]. Likewise, it has been shown that 40% loss of dopaminergic cells in the SNc in rodents is accompanied by locomotor abnormalities and a shift of STN neuronal activity from a regular to a bursty phenotype[36]. Although, 40% loss of TH-positive cells in the SNc is sufficient to observe some motor abnormalities, it constitutes a mild model of PD. Parkinson-like conditions are reached after loss of at least 70% dopaminergic neurons[37], so next we sought to validate mDBS in a severe hemiparkinsonian mouse model, which showed 90% loss of TH-positive cells in the SNc. In the hemiparkinsonian mouse model, we did not find any effects of mDBS on general locomotor activity in the open field. Because of the severity of the model, it might be possible that the open field is not sensitive enough to detect any changes in general locomotor activity following mDBS. Therefore, we also performed the rotarod and the CatWalk, which measure balance, coordination, and gait, respectively. In both tests, we found that STN mDBS improved motor symptoms substantially when comparing AMF on and off conditions in the same animal, in particular their coordination, balance, posture, and gait. When considering classical DBS, unilateral STN stimulation has also shown to effectively reverse the forelimb use asymmetry caused

by 6-OHDA injection[38], improve treadmill walking[39], performance in stepping and rotarod[40], and enhance the speed of locomotion in the CatWalk test[41]. Despite the current application of STN DBS as treatment of PD, the mechanisms underlying its symptom-alleviating effects are still not entirely understood.

Instead, data from animal models and human patients have demonstrated that classical DBS in PD exerts its therapeutic effects through multiple complex mechanisms. The majority of studies have shown that high-frequency STN DBS induces a functional inactivation of targeted structures by depolarization block mechanism and/or by stimulation of axons (for example, of GABAergic afferent neurons)[42,43]. Fibers modulated by DBS may in turn influence the physiology of brain regions at a distance from the original stimulation site. For example, with a predominance of glutamatergic projection cells, DBS in the STN has been shown to increase cell firing in structures innervated by the nucleus[44]. Additional consequences include changes in glial activity, synaptic transmission, and the development of neuroplasticity among others[28].

While the neuromodulatory potential of mDBS appears to be similar to classical DBS, the local mechanisms of action of mDBS are less complex. Local heating of MNPs opens TRPV1 channels and allows the influx of calcium ions, thereby causing neural excitation[13]. Interestingly, Gradinaru et al.[45] showed that optogenetic excitation or inhibition of STN neurons had no therapeutic effects in parkinsonian mice. Instead, the authors demonstrated that optogenetic stimulation of afferent fibers projecting from the motor cortex to the STN was able to ameliorate motor symptoms[45]. Whether mDBS also affects afferent axons projecting to the STN still needs to be elucidated, however, when considering the transfection area of the STN, we can observe some spread to areas in close vicinity to the STN (Fig. 3F–H). It might therefore be possible, that mDBS recruits a larger neuronal population than just the STN. Understanding how mDBS results in therapeutic effects in PD will be critical to understanding not only how mDBS works, but also how to develop it further and how to apply it effectively to other neurological disorders.

## Methods

**TRPV1-lentivirus**. Although TRPV1 is naturally expressed across the mammalian nervous system[46], we designed a transgene to establish sustained and uniform levels of TRPV1 expression for magnetothermal membrane depolarization[47]. The preparation of pLenti-CaMKIIα-TRPV1-p2A-mCherry was performed by Boston Children's Hospital viral core, which yielded $10^{11}$ transducing units/ml.

**Magnetic nanoparticles**. MNPs consist of an iron-oxide core coated with polyethylene glycol-poly(maleic anhydride-alt-1-octadecene) (PEG-PMAO) polymer to improve its biocompatibility and colloidal stability[48]. They exhibit specific loss power (heating efficiency per gram of iron) of about 600 W/g, in an AMF at a therapeutically relevant frequency $f = 160$ kHz and field amplitude $H_0 = 30$ kA/m, sufficient to achieve a temperature increase of 6 °C in order to trigger reversible firing of TRPV1-expressing neurons[23]. The non-magnetic nanoparticles consist of wüstite iron-oxide and lack magnetic properties, and therefore neither dissipate heat nor activate neurons when exposed to an AMF.

**Magnetic coils**. The magnetic coils used in this study were custom built in our lab at the Department of Material Science and Engineering in MIT, USA (for reference, see ref.[24]).

**Cell culture**. Cell culture was maintained in DMEM with GlutaMAX™ supplemented with 10% fetal bovine serum (FBS). To promote adhesion of the HEK293FT cells to glass substrate for imaging during field stimulation, 5 mm glass slides were coated with 100 µl of Matrigel® solution (50 µl Matrigel® in 1.5 ml of DMEM + FBS) overnight prior to cell seeding at 80% confluency in 24-well plates. Transfections were performed using 1 µl of Lipofectamine® 2000 with 500 ng of total DNA in Opti-MEM and used for imaging and magnetothermal stimulation a day after transfection. Dual viral transfection was achieved by first adding 1 µl of the Lenti-CaMKIIα::TRPV1-p2A-mCherry followed by a 3-day induction period.

Then, 0.5 µl ($10^{12}$ transducing units/ml) of AAV9-hSyn::GCaMP6s (PennVector Core) was added. In vitro magnetothermal stimulation was performed 5 days later.

**In vitro magnetothermal stimulation**. HEK293FT cells were shortly incubated in either Tyrode (control) or 2 mg/ml ferrofluid in Tyrode and stimulated for 20 s with AMF at 160 kHz and 30 kA/m during 60 s long fluorescence recordings (no field 0–20 s, AMF 20–40s, no field after 40 s). In total, 30 cells per condition were analyzed using custom-written Mathematica and Matlab codes based on work from Gregurec et al.[49].

**In vivo virus and nanoparticle injection**. The in vivo study consisted of three experiments:

Experiment 1: Adult male C57BL/6 mice (Jackson Laboratory) aged 8 weeks were group-housed in standard ventilated cages (IVC) in a controlled environment (temperature 22 °C, humidity 59 (rH) using a 12/12-h dark/light cycle (light on 7 AM–7 PM). Food and water were given ad libitum. All experiments were carried out and approved by the Massachusetts Institute of Technology Committee on Animal Care (protocol # 0713-063-16).

Mice were randomly assigned to either STN mDBS ($n = 7$) or STN controls ($n = 10$).

All mice underwent isoflurane anesthesia and were placed into a stereotactic frame. Body temperature was kept at 37 °C using a thermoregulated heating pad. Consecutively, a burr hole above the left STN (AP: −2.06 mm, ML: −1.50 mm, DV: −4.50) was made and a total of 1.5 µl of Lenti-CaMKIIα::TRPV1-p2A-mCherry were injected with a microinjection apparatus (10 µl Nanofil Syringe, bevelled 34-gauge needles, UMP-3 syringe pump, and its controller Micro4 (World Precision Instruments)). The infusion rate for intracranial virus injection was 100 nl/min. Once 750 nl were injected, the syringe was elevated by 250 µm from the initial coordinates. After the injections, the syringe needle remained inside the brain for another 10 min prior to a slow withdrawal.

Following a 6–8-week incubation period for channel/reporter expression, a second injection delivered MNPs (2 µl of 80 mg/ml) or non-MNPs using the same coordinates of the STN at an infusion rate of 60 nl/min (Fig. 2A). Mice injected with non-magnetic nanoparticles served as a control group.

Experiment 2: In the second experiment we used 40 male wild-type mice (C57BL/6, Jackson Laboratory), group-housed in standard ventilated cages (IVC) in a controlled environment (temperature 22 °C, humidity 59 (rH) using a 12/12-h reversed dark/light cycle (lights off 7 AM–7 PM). Food and water were given ad libitum. Animal procedures in experiment 2 were carried out under a protocol approved by the Institutional Animal Care Committee of Maastricht University in accordance to the Central Authority for Scientific Procedures on Animals (CCD; protocol # AVD1070020186046).

The stereotactic procedure is the same as mentioned above. This time, however, we injected Lenti-CaMKIIα::TRPV1-p2A-mCherry bilaterally into the STN (1.5 µl per hemisphere). Seven weeks later, 2 µl of 80 mg/ml MNPs (SMG-25, Ocean Nanotech, San Diego, USA) or non-MNPs were also injected bilaterally into the STN. Following a 1-week recovery period, MPTP (18 mg/kg in saline, intraperitoneally) or vehicle was administered two times in one day, each injection 2 h apart. We wanted to employ an acute MPTP treatment with 4 injections in one day, however, ~40% of the mice were found dead after the second injection and we decided to abort the following 2 injections. The final group numbers were naive control ($n = 8$), naive mDBS ($n = 10$), MPTP control ($n = 6$), MPTP mDBS ($n = 7$). Following 2 weeks, mice were placed into the AMF coil and stimulated for 3 min. After 3 min, they were taken out of the coil and subsequently tested in the open field.

Experiment 3: In the third experiment we used 22 male wild-type mice (C57BL/6, Jackson Laboratory), group-housed in standard ventilated cages (IVC) in a controlled environment (temperature 22 °C, humidity 59 (rH) using a 12/12-h reversed dark/light cycle (light off 7 AM–7 PM). Food and water were given ad libitum. Animal procedures in experiment 3 were carried out under a protocol approved by the Institutional Animal Care Committee of Maastricht University in accordance to the CCD (protocol # AVD1070020186046).

The stereotactic procedure is the same as mentioned above. This time, however, we injected Lenti-CaMKIIα::TRPV1-p2A-mCherry unilaterally into the left STN (1.5 µl per hemisphere). Seven weeks later, 1.5 µl of 80 mg/ml MNPs (SMG-25, Ocean Nanotech, San Diego, USA) were also injected into the STN. In the same surgery, 0.2 µl of 6-OHDA was also unilaterally injected into the ipsilateral median forebrain bundle (AP: −1.2 mm, ML: −1.1 mm, DV: −5 mm) at a rate of 0.1 µl/min (3 µg total). Two mice died during surgery and we experienced an additional 50% drop-out due to poor body condition of the 6-OHDA mice. We therefore switched to a within-subjects design and were able to test $n = 10$ mice before and after AMF stimulation in the rotarod, open field, and CatWalk gait analysis system 2 weeks after surgery.

**Behavioral tests**. All behavioral tests were performed under low light conditions, and animals were allowed to acclimate to the behavior room for at least 1 h before the beginning of behavioral testing.

**Rotational behavior**. The circular arena was made out of plexiglass (Ø 9 mm, height 10 cm, fitting within the magnetic coil). Mice were placed into the arena and AMF stimulation was turned on for 3 min for both, MNPs and non-MNPs injected mice. The videos were manually scored independently by two individual researchers and rotations were classified as ipsi- or contralateral rotations around the body axis. The frequency of rotations was assessed per 3 min and per 30 s time bins.

**Open field**. The square arena of the open field consists of walls of 25 cm in height and dimensions of 50 cm width × 50 cm length. The tracking system of the open field behavior was done in a semi-dark condition for each mouse, using a tracking software (Ethovision, Noldus Information Technology, Wageningen, The Netherlands). The open field trial duration lasted 5 min and each mouse was stimulated in an AMF coil for 3 min before testing (Fig. 4A).

**Rotarod**. An accelerating rotarod with a grooved rotating beam (3 cm) raised 16 cm above a platform (model 47650, Ugo Basile, Italy) was used to measure coordination. The latency to fall off the rotating rod was recorded. Prior to 6-OHDA lesion, mice were pre-trained for 2 days in order to reach a stable performance. The training consisted of 3 sessions per day on 2 consecutive days, whereby each session lasted 120 s. On day 1, mice were trained at 5 rpm and on day 2, at 10 rpm. The final test (three sessions, each lasting 300 s) was performed after 6-OHDA lesion with and without AMF stimulation with 1 week in between the two conditions. Testing was done with speeds starting at 4 rpm and accelerating to 40 rpm within 300 s. Between trials, mice were given at least 2 min of rest in order to reduce stress and fatigue. Values are expressed as mean latency to fall from the rotarod in the 3 test trials.

**CatWalk XT**. Mice were subjected to gait assessment with the CatWalk-automated gait analysis system (Noldus Information Technology, Wageningen, The Netherlands) before and after AMF stimulation with 1 week in between. The apparatus comprises a long glass plate with a fluorescent light beamed into the glass walkway floor from one side. In a dim environment, the light is reflected downward and the footprints of the mouse as it walks along the walkway are recorded by a camera mounted under the glass. The glass plate was cleaned and dried before testing each subject to minimize the transmission of olfactory cues. In general, one successful test recording consisted of an average of three uninterrupted runs having a comparable running speed with a maximum variation of 60%. The following general, coordination, static and dynamic parameters assessing individual paw functioning and gait patterns were analyzed: run duration, number of steps, number of step patterns, step pattern regularity index (%), stride length, max contact area, print area, print length, print width, initial dual stance, terminal dual stance and body speed variation, three limb support and diagonal limb support.

**Tissue collection**. At the end of the experiments, the mice were overdosed with pentobarbital. Transcardial perfusions with PBS and then 4% paraformaldehyde were carried out. Brains were removed and placed in fresh fixative overnight at 4 °C. Subsequently, brains were transferred to 1% NaN₃ at 4 °C for long-term storage.

For vibratome sectioning (Leica®, Wetzlar, Germany), brains were embedded in 10% gelatin from porcine skin (Sigma-Aldrich, Zwijndrecht, The Netherlands), and then cut into 30 μm slices in the frontal plane on a vibratome (Leica®, Wetzlar, Germany). Slices were immediately transferred into 1% NaN₃ and kept at 4 °C.

**Immunohistochemistry**. For immunohistochemistry, sections were incubated overnight with polyclonal rabbit anti-c-Fos primary antibody (1:1000; K-25, Santa Cruz Biotechnology Inc, Santa Cruz, USA) or rabbit anti-tyrosine hydroxylase (TH, 1:500, H-196, Santa Cruz Biotechnology Inc, Santa Cruz, USA). C-Fos immunohistochemistry included biotinylated donkey anti-rabbit secondary antibody (1:400; Jackson Immunoresearch Laboratories Inc., Westgrove, USA) and avidin–biotin peroxidase complex (1:800, Elite ABC-kit, Vectastain®, Burlingame, CA, USA). The staining was visualized by 3,3′-Diaminobenzidine (DAB) combined with NiCl₂ intensification. TH was visualized using immunofluorescence with donkey anti-rabbit Alexa 488 secondary antibody (1:200, Invitrogen, Carlsbad, CA, USA). The number of c-Fos and TH-positive cells was counted using the stereological procedure, optical fractionator. Counts were done using a microscope (Olympus® BX51W1), a motorized stage, and the StereoInvestigator software (MicroBrightField, Williston, VT). All c-Fos and TH-positive cells in an average of 2–4 sections per region of interest, 300 μm apart were counted with a 40× objective. The total number of positive cells was estimated as a function of the number of cells counted and the sampling probability.

**Statistical analysis**. All data are represented as mean ± standard error of the mean (SEM), and analyses were performed with SPSS Statistics (version 25) or Graph Pad Prism (version 8). Normality and homogeneity of variance of the data were checked using the Shapiro–Wilk test and normality plots. The in-vitro data was analyzed by repeated-measures ANOVA, followed by a Bonferroni post hoc test. Since we did not have repeated experiments, we also performed a Tukey analysis.

In order to stabilize variance in the rotational behavior between the groups, we performed a log transformation. Next, we performed a repeated-measures ANOVA from 30 to 180 s, when MNPs reached their peak effect. Following this, we performed a one-tailed $t$-test to determine which time points showed statistical significance. The total rotations were also analyzed by a one-tailed $t$-test. The open field data of the MPTP experiment was analyzed using a one-way ANOVA followed by Fisher's least significance post hoc test. All behavioral data of the 6-OHDA experiment were analyzed using within-subjects repeated-measures ANOVA comparing before and after AMF stimulation. Stereological cell counts were analyzed by independent samples $t$-test. $P$-values < 0.05 were considered significant.

**Reporting summary**. Further information on research design is available in the Nature Research Reporting Summary linked to this article.

## Data availability

All data are available within the manuscript, figures, and tables, and have been deposited in the GitHub database [https://github.com/shescham/magnetothermalDBS.git]. Individual data points are shown in all figures. Further information and request for reagents will be available upon reasonable request to the corresponding author. Source data are provided with this paper.

## Code availability

Analysis scripts are available at https://github.com/shescham/magnetothermalDBS.git.

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

## Acknowledgements

We thank F. Zhang (MIT) for sharing the HEK293FT cell line, D. Julius (UCSF) for sharing Rat Vr1 (TRPV1) in pCDNA3, and K. Deisseroth (Stanford) for sharing pLenti-CaMKIIα-hChR2 (T159C)-p2A-mCherry-WPRE. We also thank H. Duimel, C. López-Iglesias, and the Microscopy CORE Lab of M4I-FHML, Maastricht University, for their help with processing brains for TEM imaging. We also wish to thank A. Blokland for his help in conducting the statistical analysis, and M. Roet and S. Rao for their support throughout various aspects of the study. This work was partly funded by the NIH BRAIN Initiative grant (1R01MH111872) which was awarded to A.P. and P.A.; S.H. is a recipient of the NWO VENI Fellowship; J.M. is a recipient of the Samsung scholarship; M.G.C. is a recipient of the ETH Zurich Postdoctoral Fellowship; H.L. is funded by the Chinese Scholarship Council.

## Author contributions

S.H., A.P., P.A. and Y.T. designed the experiments and wrote the paper; D.G. and J.M. synthesized, functionalized, and characterized the MNPs; D.G. and J.M. conducted the in vitro work and carried out data analysis; M.G.C. designed the AMF coils for in vitro and in vivo experiments; S.H. carried out data collection and analysis for all in vivo experiments; P.C. performed the surgeries of in vivo experiment 1; A.J. and H.L. participated in MPTP experiments; H.L. performed stereological cell counting; D.R. checked the location of injection and created injection maps; P.A. and Y.T. supervised all aspects of the work.

## Competing interests

The authors declare no competing interests.
