## [Peer Review File · Nature Communications]

Reviewers' Comments:

Reviewer #1:

Remarks to the Author:

In this study, Heschem et al. examine the effect of the magnetothermal stimulation in MPTP (parkinsonian) mice as a non-invasive alternative for electrical deep brain stimulation (DBS) of the subthalamic nucleus (STN) in parkinsonian patients. The authors have validated this powerful technique, the magnetothermal stimulation, in an impressive and convincing set of previous publications (for example Munshi et al., eLife 2017; Chen et al., 2015 Science).

The authors first verified the effect of the magnetothermal stimulation in HEK cells transfected with TRPV1, which is a ionotropic receptor opening upon an increase of the membrane temperature, via excitation of magnetic nanoparticules (MNPs, brought by antibodies) by magnetic stimulation. They test the behavioral effect of magnetic stimulation in mice with TRPV1 and MNP colocalized in neurons of the STN in "control" mice, and next in MPTP-treated mice (used as parkinsonian rodent model). The authors conclude that magnetic DBS (mDBS) of STN neurons is able to rescue some motor defects (distance moved) in parkinsonian mice without significant effect in the various control mice.

Overall, this study provides a very interesting and promising tool to manipulate neuronal population and putatively to propose new therapeutic alternative for DBS. However, there are some loose ends which need to be tied up. Despite elegant sets of experiments and the application of a very sophisticated technique, some questions remain open and some experiments do not fully support the very strong conclusions brought by the authors about the ability of mDBS to alleviate some motor defect in MPTP mice.

Major comments:

- 1) It would be great to have the precise location of the injection and of the transfected cells in the targeted structure (STN); the authors could provide a summary graph illustrating the different areas of transfected cells for example. The expression of TRPV1 can be visualized with the fluorescence reporter (maybe the same for MNPs?). It is important that the authors show the specific (or not) expression of TRPV1 (and MNPs if possible) in the STN. This point will be also very helpful to discuss the distinct effects reported here when compared to the opto-stimulation of STN neurons (Gradinaru et al., 2009). Indeed, considering that only STN neurons are transfected with TRPV1 (and MNPs), how the authors explain the discrepancy between their results and those of Gradinaru's paper? Having different results is not a problem by itself, but it needs to be explained. Maybe an explanation could be that the TRPV1/MNPs transfection affect larger neuronal population than just the STN. Indeed it has been described that electrical DBS should recruit larger neuronal population than just the STN to be fully efficient.
- 2) It is not clear how many mice were tested for the data illustrated in Figures 2 and 3. Please give this information in the Results and Figure Legends.
- 3) The authors should provide information about the effect of the overexpression of TRPV1 in STN neurons. Do the passive and active membrane properties, firing rate or PPR affected? This can be achieved by patch-clamp recordings in acute brain slices of from transfected mice. Indeed, when compared to optogenetics for which in absence of opto-stimulation, the opsins remain closed, TRPV1 could be still activated by endogenous ligands, such as anandamide. In this line, what is known about endocannabinoid system in STN?
- 4) Figure 3 and c-Fos expression: the authors should explain why c-Fos positive cells are quite numerous in the primary motor cortex in control conditions (which less/not the case in the red nucleus or amygdala). Ideally, it would have been great to get the comparison of the number of c-Fos positive cells following electrical DBS. I am not necessarily requesting such (time consuming and challenging) experiments, but the authors should at least discuss this point based on data available in the literature (performed in rats for most of the papers).
- 5) A main concern here is the use of a parkinsonian rodent model inducing just 40% of TH-positive cell death in SNc. Loss of 40% of SNc dopaminergic neuron can be eventually sufficient to observe

some motor abnormalities, but certainly not to be in parkinsonian-like conditions, reached after loss of 70% (at least) of dopaminergic cell. It will be required before firmly conclude to provide evidences of the efficiency of mDBS using classical parkinsonian rodent models inducing >70% of dopaminergic neurons; it is classically admitted that below 65-70% the main parkinsonian motor symptoms cannot be observed.

The authors can use the MPTP model (but reaching a higher rate of TH-positive cell death) and/or the 6-OHDA lesioned mice. Unilateral 6-OHDA-lesioned mice enable testing easily many asymmetrical motor behaviors. Indeed, assessing only the distance moved is much too weak to firmly conclude about the efficiency of mDBS to alleviate parkinsonian symptoms (at least 2-3 distinct motor behavioral tasks/measurements are required).

5) The authors should remove the last sentence of the abstract or at least tune it down. Indeed, TRPV1 and MNPs have to be injected into deep structures (STN) at the same position than the DBS electrode bringing the same risks and limitations for a therapeutic use. In addition, the effects of mDBS seem to last for 2.5 minutes after the offset of mDBS, implying than for full effects as those obtained with DBS, mDBS should be applied constantly with the need for a portable coil on the animal/patient. This should be clearly stated and discussed.

In the same line, the title should be rewritten to bring more careful conclusions. The main problem of the electrical DBS (as mentioned by the authors) is to be very invasive and to benefit only to 10% of the Parkinsonian patients. On this aspect mDBS could be a very promising alternative. Nevertheless, at this day mDBS requires to inject in deep structures TRPV1 constructs and MNPs (two distinct injections into STN), which is at the end quite similar, in term of being invasive to DBS. This point should be also discussed.

Minor comments:

1) Introduction (l62). This sentence should be in the last paragraph (from l73) since it concerns the current study. Also, please detailed a bit more about the principles (and associated refs) of the magnetothermal stimulation; this is a great and promising technique, which deserves more detailed information.

2) Figure 1C left panel: in most illustrated cells one can observe a calcium increase during the OFF period (which is not the case, as expected, in the right panel); why such calcium increase? In addition, it is not clear if this is the representation of a single experiment (many cells being monitored during a single experiment)? Or were these 30 cells monitored from many experiments? The authors need to clearly indicate how many independent experiments were performed and provide the overall effect. Please clarify this.

Reviewer #2:

Remarks to the Author:

In this manuscript, Dr. Temel and colleagues describe an innovative new approach for stimulating neural tissue wirelessly in a mouse model. Specifically, they develop a magnetothermal approach that activates heat-sensitive ion channels in neural tissue using TRPV1. Using this approach, they provide convincing evidence of clinical effect in a PD mouse model. They also provide well designed controls in which the same viral delivery is provided but in which nonmagnetic particles are infused. Overall, this approach is noteworthy because it could provide an important new method for delivering DBS in patients with movement or other neurocognitive disorders.

While the experiments are well design and the topic important, there are a number of points that would need to be addressed.

First, regarding novelty, there is a fairly long history of using analogous approaches including use of gold nanoparticles (without heat-sensitive ion channel-based control) for modulating neural tissue activity. These include, among others, papers by Johannsmeier et al Scientific Reports 2018 and Zhang et al Nano Letters 2013. Ed Boyden also has a really nice paper in Cell 2017 using temporally interfering electromagnetic waves. These and similar references need to be noted and

discussed in more detail (but sorry if I missed these).

Second, the authors need to provide detailed information about the depth and breadth of stimulation. While they provide evidence of cfos activation, it would be helpful to provide a formal statistical description of the stimulation drop-off/penumbra.

Third, while these experiments are done in mice, magnetic field strength drop-off exponentially with distance. This is not a big issue with mice but can become a major challenge in humans. It would be critical to demonstrate how effective such stimulation may be at larger distances from target (e.g., 10-20 cm). It would also be important to confirm that there is no tissue damage.

Fourth, it would be important to quantify the duration of effect over weeks-to-months. A critical benefit of standard DBS is its longevity. It is unclear, however, how long the effect of TRPV1-based stimulation lasts after nanoparticle implantation. Although the present paper provides a nice proof of concept, its major stated goal is to offer a potential substitution/complementation to standard clinical DBS. Such long-term studies would therefore be necessary.

Reviewer #3:

Remarks to the Author:

This manuscript describes in vitro and in vivo magnetothermal deep brain stimulation (mDBS) for treating Parkinson's disease. The topic is very intriguing and worth investigating. I recommend publishing this manuscript in Nature Communications after minor revisions.

Specifically,

- Could the authors provide biocompatibility test and cell viability data of both MNPs and the non-magnetic FeO particles?
- For the statistical analysis, why was one-tailed t-test conducted instead of two-tailed t-test?
- Line 112, what is the meaning of (80, 10440)? The ANOVA, while linked to Figure 1C, there is no figure panel showing the statistical analysis. Besides, the bottom plots and the heat maps may not be consistent in the dF/F_0 .
- Line 113 and 417, should it be dF/F_0 , instead of dF_0/F ?
- Line 116, while the authors stated that different frequency and amplitude was used in previous example, can the author provide the optimization and rationale of using various experimental conditions?
- Line 127 and figure 2A, while saying "[...], which was followed by MNP injection into the same region 4 weeks later (Fig. 2A)", the figure labelled "6-8 weeks" for the time point.
- Line 158 and figure 4A, similar problem as above. The figure shows "7 weeks" time point while 4 weeks stated in the text.
- Figure 3B, the staining of c-Fos (K-25) are not obvious.

Response to reviewers:

We appreciate the reviewers' perspective on our technology and believe that the additional experiments and discussion prompted by their comments substantially strengthened our manuscript. To address all of the reviewers' concerns, we performed additional experiments and expanded our discussion to further clarify our approaches and the potential future applications and limitations. Our responses (denoted in blue) to the specific reviewers' comments are provided below.

Reviewer # 1

In this study, Heschem et al. examine the effect of the magnetothermal stimulation in MPTP (parkinsonian) mice as a non-invasive alternative for electrical deep brain stimulation (DBS) of the subthalamic nucleus (STN) in parkinsonian patients. The authors have validated this powerful technique, the magnetothermal stimulation, in an impressive and convincing set of previous publications (for example Munshi et al., eLife 2017; Chen et al., 2015 Science).

The authors first verified the effect of the magnetothermal stimulation in HEK cells transfected with TRPV1, which is a ionotropic receptor opening upon an increase of the membrane temperature, via excitation of magnetic nanoparticules (MNPs, brought by antibodies) by magnetic stimulation. They test the behavioral effect of magnetic stimulation in mice with TRPV1 and MNP colocalized in neurons of the STN in "control" mice, and next in MPTP-treated mice (used as parkinsonian rodent model). The authors conclude that magnetic DBS (mDBS) of STN neurons is able to rescue some motor defects (distance moved) in parkinsonian mice without significant effect in the various control mice.

Overall, this study provides a very interesting and promising tool to manipulate neuronal population and putatively to propose new therapeutic alternative for DBS. However, there are some loose ends which need to be tied up. Despite elegant sets of experiments and the application of a very sophisticated technique, some questions remain open and some experiments do not fully support the very strong conclusions brought by the authors about the ability of mDBS to alleviate some motor defect in MPTP mice.

Major comments:

Comment 1) It would be great to have the precise location of the injection and of the transfected cells in the targeted structure (STN); the authors could provide a summary graph illustrating the different areas of transfected cells for example. The expression of TRPV1 can be visualized with the fluorescence reporter (maybe the same for MNPs?). It is important that the authors show the specific (or not) expression of TRPV1 (and MNPs if possible) in the STN.

This point will be also very helpful to discuss the distinct effects reported here when compared to the opto-stimulation of STN neurons (Gradinaru et al., 2009). Indeed, considering that only STN neurons are transfected with TRPV1 (and MNPs), how the authors explain the discrepancy between their results and those of Gradinaru's paper? Having different results is not a problem by itself, but it needs to be explained. Maybe an explanation could be that the TRPV1/MNPs transfection affect larger neuronal population than just the STN. Indeed it has been described that electrical DBS should recruit larger neuronal population than just the STN to be fully efficient.

Response: We thank the reviewer for this comment and have included a representative picture of the precise location of the MNPs injection and of the transfected cells in the STN to the revised Figure 3 of the manuscript.

Figure 3: (F and G) Injection maps of (F) MNPs and (G) TRPV1 in the brain (n = 10 mice) and demonstrating the coverage areas within the STN (marked in black). (H) Percentage of area covered by MNPs, TRPV1 or both in the STN.

Indeed, Gradinaru et al. (2009) showed that optogenetic excitation or inhibition of STN neurons had no therapeutic effects in parkinsonian mice. Instead, the authors demonstrated that optogenetic stimulation of afferent fibers projecting from the motor cortex to the STN was able to ameliorate motor symptoms. When considering the transfection area of the STN, we can observe some spread to areas in close vicinity to the STN (Fig. 3F and G). It might therefore be well possible, that mDBS recruits a larger neuronal population than just the STN. A statistical description of the stimulation drop-off/penumbra has been added to the supplementary materials (Fig. S2).

Comment 2) It is not clear how many mice were tested for the data illustrated in Figures 2 and 3. Please give this information in the Results and Figure Legends.

Response: We apologize for not clarifying the exact numbers of animals used in the present study. We have now included the number of mice that were tested for the data illustrated in Figures 2 and 3 in the revised version of the manuscript. Mice tested for the data in Figure 2 were n = 7 for STN mDBS and n = 10 served as control mice. From these mice we then selected n = 7 for STN mDBS and n = 6 as control mice to perform c-Fos immunohistochemistry.

Comment 3) The authors should provide information about the effect of the overexpression of TRPV1 in STN neurons. Do the passive and active membrane properties, firing rate or PPR affected? This can be achieved by patch-clamp recordings in acute brain slices of from transfected mice.

Indeed, when compared to optogenetics for which in absence of opto-stimulation, the opsins remain closed, TRPV1 could be still activated by endogenous ligands, such as anandamide.

In this line, what is known about endocannabinoid system in STN?

Response: We thank the reviewer for his suggestion. In fact, the expression and function of TRPV1 in the CNS is still controversial. Since patch-clamp recordings in acute brain slices would go beyond the scope of this study, we have added the following to the discussion of our revised version of the manuscript: “Here, we targeted the STN with mDBS and assessed its biomedical potential as wireless alternative to classical DBS. Although, there are precedents of applying nanomaterials-mediated neuromodulation in clinically-inspired contexts, including photothermal approaches using gold nanoparticles (Johannsmeier et al., 2018; H. Zhang, Shih, Zhu, & Kotov, 2012), there have been no prior reports of translational studies aiming to treat parkinsonian symptoms using mDBS. Our study relied on exogenously expressed TRPV1 to create uniform levels across mice, which might represent a major challenge for clinical applications. This channel, however, is also endogenously expressed in neurons and glia in certain regions of the mammalian central nervous system (Roet, Jansen, Hoogland, Temel, & Jahanshahi, 2019). Various studies have shown that TRPV1 can be found in prefrontal cortex, amygdala, hypothalamus, periaqueductal gray, locus coeruleus, cerebellum, hippocampus and

dentate gyrus (Chávez, Chiu, & Castillo, 2010). Whether TRPV1 is also endogenously expressed in the STN, is yet to be elucidated. Notably, TRPV1 receptors can mediate a presynaptic form of long-term depression (LTD) at glutamatergic synapses onto CA1 inhibitory interneurons in the hippocampus (Chávez et al., 2010). In this respect, endogenous lipid ligands such as anandamide (AEA), can trigger a form of postsynaptic LTD. In the present study, we did not observe any abnormal behaviors in mice transduced to overexpress TRPV1 in the STN neurons. Moreover, control mice followed the same surgical procedure including an injection of a TRPV1-carrying vector. Nevertheless, to rule out possible side effects caused by endogenous AEA, future studies could generate AEA insensitive TRPV1 mutants. Another point of consideration with regard to clinical application is the use of AMF coils. Magnetic coils suitable for cell-destructive therapy in magnetic hyperthermia and magnetothermal neuromodulation in rodents can be engineered to efficiently generate appropriate AMF conditions temporarily over small experimental volumes (Christiansen, Howe, Bono, Perreault, & Anikeeva, 2017; Munshi et al., 2017). For chronic use, however, a portable coil on the animal/patient is necessary to maintain therapeutic benefits. In line with this, scaling AMF coils to volumes necessary for neuromodulation in human patients also present a formidable challenge, as the power requirements to achieve comparable AMF conditions increase substantially (Christiansen et al., 2017; Lacroix, Carrey, & Respaud, 2008).”

Comment 4) Figure 3 and c-Fos expression: the authors should explain why c-Fos positive cells are quite numerous in the primary motor cortex in control conditions (which less/not the case in the red nucleus or amygdala).

Ideally, it would have been great to get the comparison of the number of c-Fos positive cells following electrical DBS. I am not necessarily requesting such (time consuming and challenging) experiments, but the authors should at least discuss this point based on data available in the literature (performed in rats for most of the papers).

Response: For the c-Fos immunohistochemistry we have selected 2 motor regions, the primary motor cortex and the red nucleus, as well as one non-motor region such as the amygdala.

It has been shown before that locomotion coincides with c-Fos expression. In the red nucleus, there is considerable functional redundancy between the corticospinal tract and the rubrospinal tract. Therefore, in case of corticospinal tract injury (such as stroke or spinal cord injury), the red nuclei may provide a small degree of compensation in motor function and thus increase their neural activity. The present study, however, included healthy wild-type mice and therefore the corticospinal tract functions as primary motor control unit. This might be a reason why fewer c-Fos cells are observed in the red nucleus when compared to the primary motor cortex. At the time of experiments the animals also did not experience anxiety and were handled carefully, therefore neural activity in the amygdala is scarce in both groups (mDBS and control).

With regard to the high levels of c-Fos in the motor cortex, it is generally believed that when a neuron receives strong and sustained glutamatergic excitatory inputs along with GABA inhibitory modulation, c-Fos expression is rapidly turned on. The striatum and STN are input stations of the basal ganglia and receive inputs from a wide area of the motor cortex. The information is processed through the hyperdirect, direct, and indirect pathways and reaches the GPi/SNr, the output station of the basal ganglia. During voluntary movements, neuronal signals originating in the cortex are considered to be transmitted through these pathways, reach the GPi/SNr and control movements. Signal transmission through the direct pathway reduces GPi activity and facilitates movements by disinhibiting the thalamus, whereas the hyperdirect and indirect pathways increase GPi activity and suppress movements. STN-DBS increases activity of GPi neurons through the excitatory STN-GPi projections. Consequently, less GABA is released to the thalamus which in turn activates the motor cortex and alleviates symptoms of bradykinesia.

Comment 5) A main concern here is the use of a parkinsonian rodent model inducing just 40% of TH-positive cell death in SNc. Loss of 40% of SNc dopaminergic neuron can be eventually sufficient to observe some motor abnormalities, but certainly not to be in parkinsonian-like conditions, reached after loss of 70% (at least) of dopaminergic cell. It will be required before firmly conclude to provide evidences of the efficiency of mDBS using classical parkinsonian rodent models inducing >70% of dopaminergic neurons; it is classically admitted that below 65-70% the main parkinsonian motor symptoms cannot be observed.

The authors can use the MPTP model (but reaching a higher rate of TH-positive cell death) and/or the 6-OHDA lesioned mice. Unilateral 6-OHDA-lesioned mice enable testing easily many asymmetrical motor behaviors. Indeed, assessing only the distance moved is much too weak to firmly conclude about the efficiency of mDBS to alleviate parkinsonian symptoms (at least 2-3 distinct motor behavioral tasks/measurements are required).

Response: We thank the reviewer for his suggestion. It is true that in PD patients first symptoms appear when circa 70% DA cell are lost. Such a late manifestation of symptoms must be due to the existence of compensatory mechanisms. That being said, no clear histological data are available for patients with mild symptomatology, since post-mortem brain tissue mainly stems from patients who died at an advanced stage of PD. The discrepancy between rodent models and human models in the amount of DA loss and symptoms is a known one. In line with this, we have previously shown that animals with a 40% loss of TH cells in the SNc were slower in the motor response of a reaction time task, and electrophysiological properties of the STN firing pattern already shifted to a bursty phenotype, which is also present in severe PD models (Janssen et al., 2012). We do admit, however, that 40% TH cell loss in the SNc is generally considered as the onset of PD/mild PD and have therefore tuned down some of the statements in the discussion. We have also added the following to the revised version of the manuscript: “A general decline in motor frequency or ability to initiate movement comprises bradykinesia, a cardinal symptom of PD. Loss of TH at bradykinesia onset in PD is ~80% in striatum and 40% in the SNc, with similar loss estimated in PD patients (Salvatore, McInnis, Cantu, Apple, & Pruett, 2019). Likewise, it has been shown that 40% loss of dopaminergic cells in the SNc in rodents is accompanied by locomotor abnormalities and a shift of STN neuronal activity from a regular to a bursty phenotype (Janssen et al., 2012).”

Additionally, we have now added a new set of experiments aimed at investigating the effects of mDBS in a severe hemiparkinsonian model using unilateral 6-OHDA-lesioned mice, which showed 90% loss of TH-positive cells in the SNc. Mice were tested in the Open Field, Rotarod and Catwalk. In the hemiparkinsonian mouse model, we did not find any effects of mDBS on general locomotor activity in the open field. Because of the severity of the model, it might be possible that the open field is not sensitive enough to detect any changes in general locomotor activity following mDBS. Therefore, we also performed the rotarod and the Catwalk, which measure balance, coordination and gait, respectively. In both tests, we found that STN mDBS improved motor symptoms substantially when comparing AMF on and off conditions in the same animal, in particular their coordination, balance, posture and gait (see figure 4 in the revised version of the manuscript).

Fig 4: (A) In-vivo experimental scheme for the translational study. (B) Representative photomicrographs of TH-positive cells in the substantia nigra pars compacta contralateral and ipsilateral to the 6-OHDA lesion. Scale bar 50 μ m. (C) Total distance moved (in cm) in the Open Field \pm S.E.M. There was no significant difference between AMF off and AMF on conditions when considering the first 2.5 min. (D) Latency to fall from the rotarod (in s) during AMF off and AMF on conditions averaged from 3 consecutive trials. Mice that were stimulated in an AMF 3 min before the rotarod, significantly improved their motor performance. (E-L) Representative CatWalk results. AMF stimulation resulted in faster run duration (E), less number of steps (F), increased step sequence regularity (%) (G), increased diagonal limb support (H), increased stride length (I), decreased paw print area (J), decreased initial dual stance (K), and decreased body speed variation (L). Abbreviations: RF: right front paw, RH: right hind paw, LF: left front paw, LH: left hind paw.

Comment 6) The authors should remove the last sentence of the abstract or at least tune it down. Indeed, TRPV1 and MNPs have to be injected into deep structures (STN) at the same position than the DBS electrode bringing the same risks and limitations for a therapeutic use. In addition, the effects of mDBS seem to last for 2.5 minutes after the offset of mDBS, implying than for full effects as those obtained with DBS, mDBS should be applied constantly with the need for a portable coil on the animal/patient. This should be clearly stated and discussed. In the same line, the title should be rewritten to bring more careful conclusions. The main problem of the electrical DBS (as mentioned by the authors) is to be very invasive and to benefit only to 10% of the Parkinsonian patients. On this aspect mDBS could be a very promising alternative. Nevertheless, at this day mDBS requires to inject in deep structures TRPV1 constructs and MNPs (two distinct injections into STN), which is at the end quite similar, in term of being invasive to DBS. This point should be also discussed.

Response: We have removed “therapeutically” from the last sentence of the abstract and have changed the title. Moreover, we have discussed potential shortcomings, such as the practical use of mDBS with a portable coil on the patient’s head, as well as the need to inject viral constructs, in more detail in the discussion. The following section was added to the discussion:

“Here, we targeted the STN with mDBS and assessed its biomedical potential as wireless alternative to classical DBS. Although, there are precedents of applying nanomaterials-mediated neuromodulation in clinically-inspired contexts, including photothermal approaches using gold nanoparticles (Johannsmeier et al, 2018; Zhang et al, 2013), there have been no prior reports of translational studies aiming to treat parkinsonian symptoms using mDBS. Our study relied on exogenously expressed TRPV1 to create uniform levels across mice, which might represent a major challenge for clinical applications. This channel, however, is also endogenously expressed in neurons and glia in certain regions of the mammalian central nervous system (Roet et al., 2019b). Various studies have shown that TRPV1 can be found in prefrontal cortex, amygdala, hypothalamus, periaqueductal gray, locus coeruleus, cerebellum, hippocampus and dentate gyrus (Chávez et al., 2010). Whether TRPV1 is also endogenously expressed in the STN, is yet to be elucidated. Notably, TRPV1 receptors can mediate a presynaptic form of long-term depression (LTD) at glutamatergic synapses onto CA1 inhibitory interneurons in the hippocampus (Chávez et al., 2010). In this respect, endogenous lipid ligands such as anandamide (AEA), can trigger a form of postsynaptic LTD. In the present study, we did not observe any abnormal behaviors in mice transduced to overexpress TRPV1 in the STN neurons. Moreover, control mice followed the same surgical procedure including an injection of a TRPV1-carrying vector. Nevertheless, to rule out possible side effects caused by endogenous AEA, future studies could generate AEA insensitive TRPV1 mutants. Another point of consideration with regard to clinical application is the use of AMF coils. Magnetic coils suitable for cell-destructive therapy in magnetic hyperthermia and magnetothermal neuromodulation in rodents can be engineered to efficiently generate appropriate AMF conditions temporarily over small experimental volumes (Christiansen et al., 2017, Munshi et al., 2017). For chronic use, however, a portable coil on the animal/patient is necessary to maintain therapeutic benefits. In line with this, scaling AMF coils to volumes necessary for neuromodulation in human patients also present a formidable challenge, as the

power requirements to achieve comparable AMF conditions increase substantially (Christiansen et al., 2017, Lacroix et al., 2008).

Minor comments:

Comment 1) Introduction (l62). This sentence should be in the last paragraph (from l73) since it concerns the current study. Also, please detailed a bit more about the principles (and associated refs) of the magnetothermal stimulation; this is a great and promising technique, which deserves more detailed information.

Response: We have removed the sentence in line 62 and moved it to the last paragraph. In addition, we have provided more details about the principles of magnetothermal stimulation.

Comment 2) Figure 1C left panel: in most illustrated cells one can observe a calcium increase during the OFF period (which is not the case, as expected, in the right panel); why such calcium increase? In addition, it is not clear if this is the representation of a single experiment (many cells being monitored during a single experiment)? Or were these 30 cells monitored from many experiments? The authors need to clearly indicate how many independent experiments were performed and provide the overall effect. Please clarify this.

Response: The observed signal increase in the OFF period is merely background and probably some bleaching. We can observe this signal because the algorithm calculates moving averages and takes into account several seconds/frames back and forth. For our 25nm Fe₃O₄ nanoparticles, specific loss power (SLP, normalized to iron content [Fe]) is greatest when the AMF was set to 160kHz and > 30kA/m (about 600W/g, see Fig. 1B). Since we have done numerous in-vitro experiments using MNPs < 25nm that show the greatest SLP when the AMF is set to 522kHz and 15kA/m in the past (Chen, Romero, Christiansen, Mohr, & Anikeeva, 2015), here, we merely wanted to provide proof that changing the frequency to 160kHz in order to maximize the SLP for our MNPs would be in line with our previous results (for more information on AMF condition selection please see (Moon et al., 2020)). For this reason, we have conducted a single experiment, out of which 30 cells were randomly selected. The statistics are performed as within-subject instead of between-subject design.

Reviewer #2

In this manuscript, Dr. Temel and colleagues describe an innovative new approach for stimulating neural tissue wirelessly in a mouse model. Specifically, they develop a magnetothermal approach that activates heat-sensitive ion channels in neural tissue using TRPV1. Using this approach, they provide convincing evidence of clinical effect in a PD mouse model. They also provide well designed controls in which the same viral delivery is provided but in which nonmagnetic particles are infused. Overall, this approach is noteworthy because it could provide an important new method for delivering DBS in patients with movement or other neurocognitive disorders.

While the experiments are well design and the topic important, there are a number of points that would need to be addressed.

Comment 1) First, regarding novelty, there is a fairly long history of using analogous approaches including use of gold nanoparticles (without heat-sensitive ion channel-based control) for modulating neural tissue activity. These include, among others, papers by Johannsmeier et al Scientific Reports 2018 and Zhang et al Nano Letters 2013. Ed Boyden also has a really nice paper in Cell 2017 using

temporally interfering electromagnetic waves. These and similar references need to be noted and discussed in more detail (but sorry if I missed these).

Response: We thank the reviewer for his suggestion and have added a section to the discussion of our revised manuscript describing these and other studies.

Comment 2) Second, the authors need to provide detailed information about the depth and breadth of stimulation. While they provide evidence of cfos activation, it would be helpful to provide a formal statistical description of the stimulation drop-off/penumbra.

Response: We thank the reviewer for this suggestion and have added necessary information as well as the heat transfer equation to the supplementary materials. To consider the spatial extent of expected magnetothermal stimulation around the ferrofluid droplet, we employed a computational model. This model has two separate components: A) numerically solved heat transfer equations to predict the temperature expected in the vicinity of the ferrofluid droplet and B) application of a thermodynamic model of TRPV1 to predict the fraction of open TRPV1 channels that result at a given temperature. To predict the time-dependent temperature field around the droplet, we follow the approach previously described for predicting magnetothermal stimulation via bulk heating (Chen et al., 2015). Crucially, the heat sinking term of Penne's Bioheat equation is included to account for the effect of active perfusion of the tissue with blood at temperature T_b . In brief, we consider heat transfer equations in two regions: the droplet (region 1) and the surrounding tissue (region 2).

Fig. S1 Prediction of the temperature increase resulting around the injected nanoparticles subject to stimulation conditions used in this study. a) A schematic representation of the solution space considered by the partial differential equations explained in the text. Region 1 is within the spherical droplet of injected ferrofluid, whereas region 2 comprises the surrounding tissue. b) MATLAB was used to numerically solve for temperature as a function of time and radial position. The dashed line indicates the division between regions 1 and 2.

A previous experimental study of the threshold of activation for neurons overexpressing TRPV1 found a midpoint of activation at about 38 °C, with full activation at 39 °C (Munshi et al., 2017). The extent of activation depends not only on the intrinsic properties of TRPV1 but also on extrinsic circumstances including the extent of expression of and endogenous positive feedback within the neurons. Nevertheless, the temperature threshold in that work was determined for a substantially similar system and can be applied here realistically. Assuming an activation threshold of 39°C for

TRPV1 Fig S1b suggests that neurons within about 1 mm of the center of the droplet, or about half that distance from the surface of the droplet, are expected to reach this temperature during the course of stimulation. We note that neurons and brain tissue are not expected to reside within the radius of the droplet. Deviations from a spherical shape would increase the surface area to volume ratio of the droplet, increasing heat transfer efficiency and somewhat lowering temperature values inside and near the droplet. Moreover, densely concentrating MNPs can alter their effective SLP, either lowering or raising it (Deatsch & Evans, 2014).

To examine intrinsic effects on nearby TRPV1, it is possible to make use of a thermodynamic model that has been used to explain the temperature sensitivity of TRPV1 and to fit experimental characterization data (Nilius et al., 2005; Voets et al., 2004). Detailed information can be found in the supplementary materials.

To consider the influence on neurons remaining at a resting potential of -70 mV, Fig S2b shows how the calculated temperature versus time field would influence P_0 at this potential. The fractional increase in P_0 from the -70 mV 37°C base state should be proportional to the increase in ionic current permitted by TRPV1 in response to elevated temperature. To consider the range of fractional increases that might be anticipated if positive feedback played a role, the case of -70 mV (resting potential) is shown as a lower bound, and the case of depolarization at 0 mV is taken as an upper bound (Fig S2c). The role of increased temperature even somewhat further from the droplet than the 39°C threshold may be to bias them toward activation.

Fig. S2 The response of TRPV1 to temperature, in terms of the fraction of open channels P_0 is considered in the context of heat dissipated by the injected nanoparticles. a) Based on thermodynamic data from literature on TRPV1, expected P_0 versus temperature behavior is shown for several transmembrane potentials. b) Assuming a resting potential of about -70 mV, the time variation of P_0 is shown as predicted for the temperature distribution calculated in Fig S1b. c) The anticipated proportional increase in P_0 resulting from increased temperature as a function of distance from the injected droplet of magnetic nanoparticles is plotted. Because depolarization leads to increases in P_0 , positive feedback is likely and it is useful to consider the unperturbed -70 mV membrane potential as a lower bound and the depolarized case of 0 mV as an upper bound. Direct actuation is also limited to cells near enough to the injection site to be overexpress transgenic TRPV1.

Comment 3) Third, while these experiments are done in mice, magnetic field strength drop-off exponentially with distance. This is not a big issue with mice but can become a major challenge in humans. It would be critical to demonstrate how effective such stimulation may be at larger distances from target (e.g., 10-20 cm). It would also be important to confirm that there is no tissue damage.

Response: We thank the reviewer for this critique, which provides an opportunity to clarify some salient points about alternating magnetic fields (AMFs) and the devices that generate them. The reviewer's instinct is correct in anticipating the possibility for technical challenges when attempting to

scale up an AMF apparatus to a size suitable for human patients, however the actual difficulties are entirely different from the ones indicated in the comment.

In brief, the main challenges of scaling up AMFs are 1) power consumption required to generate the necessary current density 2) increased inductance of larger working volumes, and 3) adequate heat rejection. None of these limitations are fundamental, and we note that the coil used in our study here already approaches a size suitable for a human head because it had to be large enough for freely behaving mice. The challenges and design features of enlarged coils are addressed at length in a previously published paper describing a setup nearly identical to the one used in this manuscript (Christiansen et al., 2017).

Far away from coils with a net dipole moment, the field indeed drops off rapidly, with a $1/r^3$ dependence (where r is the distance from the coil) rather than an exponential dependence. However, coil geometries suitable for stimulation, such as finite solenoids or Helmholtz coils, generate a relatively homogeneous field within their working volume. The fields are not diminished appreciably by passing through tissue. We agree with the reviewer's assertion that using the fringing field from a small coil is probably not suitable for human scale coils. This is not how a realistic AMF stimulation coil would be designed.

The possibility for tissue damage is extraordinarily unlikely. Weak magnetic fields do not appreciably interact with biological tissue (Pankhurst, Connolly, Jones, & Dobson, 2003). Rapidly varying magnetic fields have two main possibilities for interaction with tissue: 1) nontargeted neuronal stimulation and 2) nonspecific heat dissipation. The frequencies used here are higher than the characteristic timescales of pulsed fields used for stimulating neurons and we did not see evidence of this kind of effect in our experiments. Nonspecific heat dissipation caused by weak eddy currents is more likely to play a role, but since the effect was based on selective heating of a ferrofluid droplet, it is merely necessary for the magnetic material to dissipate heat far more efficiently than the tissue around it. The safety limit for AMFs in cancer hyperthermia is usually stated as a field amplitude-frequency product, estimated to be 5×10^9 A/m/s (Hergt & Dutz, 2007). Since our experiments were performed at conditions below these values, we do not anticipate any nonspecific heat dissipation.

Comment 4) Fourth, it would be important to quantify the duration of effect over weeks-to-months. A critical benefit of standard DBS is its longevity. It is unclear, however, how long the effect of TRPV1-based stimulation lasts after nanoparticle implantation. Although the present paper provides a nice proof of concept, its major stated goal is to offer a potential substitution/complementation to standard clinical DBS. Such long-term studies would therefore be necessary.

Response: We thank the reviewer for pointing out this important point in the presented approach. The number of particles at the injection site could decrease via either diffusion or via endocytosis and digestion with lysosomes. Previous studies have demonstrated NP internalization into cells, mainly achieved through surface functionalization of the nanoparticles. PEG provides a bioinert hydrophilic surface coating, and various studies have shown that by adjusting the PEG layer properties, cell internalization is diminished (Suk, Xu, Kim, Hanes, & Ensign, 2016; Y. Zhang, Kohler, & Zhang, 2002). In the current study, PEGylation was used to functionalize the surface of MNPs for achieving colloidal stability in water. Besides conveying stability in physiological conditions, the PEG coating helps to minimize their internalization into the cells.

Most of the experiments presented in the manuscript were performed between 1-4 weeks post-injection of the MNPs into the STN. This demonstrates that even if there was change in the number of nanoparticles over time, it did not greatly affect the ability of the nanoparticle population to dissipate heat and evoke cell stimulation.

The following text was added to the introduction of the revised version of the manuscript: "The long-term efficiency of the magnetothermal stimulation has been recently demonstrated after two months

post-injection, suggesting a non-significant decrease of MNPs concentration at the injection site and a non-significant change to their magnetic properties (Rosenfeld et al., 2020).”

Reviewer #3

This manuscript describes in vitro and in vivo magnetothermal deep brain stimulation (mDBS) for treating Parkinson’s disease. The topic is very intriguing and worth investigating. I recommend publishing this manuscript in Nature Communications after minor revisions. Specifically:

Comment 1) Could the authors provide biocompatibility test and cell viability data of both MNPs and the non-magnetic FeO particles?

Response: We thank the reviewer for this suggestion. In fact, biocompatibility tests and cell viability data for our MNPs have been reported in our previous studies (Chen et al., 2015; Rosenfeld et al., 2020). We found that surface functionalization of MNPs with PEG yields minimal effects on cell viability as compared to other common ligands such as poly(acrylic acid) (PAA) coated MNPs (Figure provided for convenience below). In the current study, PEGylation was used to functionalize the surface of our MNPs as well as non-magnetic FeO nanoparticles for achieving colloidal stability in water. Besides conveying stability in physiological conditions, the PEG coating helps to minimize their internalization into the cells. Since both MNPs and FeO particles were coated with PEG we can assume similar biocompatibility and cell viability.

Comment 2) For the statistical analysis, why was one-tailed t-test conducted instead of two-tailed t-test?

Response: We conducted a one-tailed t-test, because we expected contralateral rotations upon unilateral STN mDBS. Similarly, unilateral electrical STN DBS induces circling behavior to the contralateral side in rodents (Meissner et al., 2002).

Comment 3) Line 112, what is the meaning of (80, 10440)? The ANOVA, while linked to Figure 1C, there is no figure panel showing the statistical analysis. Besides, the bottom plots and the heat maps may not be consistent in the dF/F0.

Response: We thank the reviewer for this observation. In line 112 (80, 10440) refers to the *degrees of freedom between* and *degrees of freedom error* of the repeated measures ANOVA, since the 60sec fluorescent recording was measured per 333ms time bin.

Since we do not have repeated experiments, we have now given values to the cells from the heatmaps. In particular, value “0” represents cells not responding to AMF and value “1” denotes the cells responding to the AMF stimulation. In Figure 1D we show the significant difference in response between cells in the presence or absence of MNPs as confirmed by Tukey analysis at $p < 0.001$ (***)

Comment 4) Line 113 and 417, should it be dF/F_0 , instead of dF_0/F ?

Response: Indeed, we have changed this.

Comment 5) Line 116, while the authors stated that different frequency and amplitude was used in previous example, can the author provide the optimization and rationale of using various experimental conditions?

Response: We apologize for not providing the rationale of using different experimental conditions in our current study. We have found that for our 25nm Fe_3O_4 nanoparticles, specific loss power (SLP, normalized to iron content [Fe]) is greatest when the AMF was set to 160kHz and $> 30kA/m$ (about 600W/g, see Fig. 1B). When using the same frequency as in our previous study (522kHz, 15kA/m) we only achieve about 250W/g. Of note, in our previous study, nanoparticles were $< 25nm$, which explains their superior performance at lower amplitude and higher frequency conditions (Moon et al., 2020). We have now described this in more detail in the revised version of our manuscript.

Comment 6) Line 127 and figure 2A, while saying “[...], which was followed by MNP injection into the same region 4 weeks later (Fig. 2A)”, the figure labelled “6-8 weeks” for the time point.

Response: Thank you for pointing this out. We have changed the text accordingly.

Comment 7) Line 158 and figure 4A, similar problem as above. The figure shows “7 weeks” time point while 4 weeks stated in the text.

Response: Thank you for pointing this out. We have changed the text accordingly.

Comment 8) Figure 3B, the staining of c-Fos (K-25) are not obvious.

Response: We have now intensified the contrast to make the staining more obvious.

References

- Chávez, A. E., Chiu, C. Q., & Castillo, P. E. (2010). TRPV1 activation by endogenous anandamide triggers postsynaptic long-term depression in dentate gyrus. *Nature neuroscience*, *13*(12), 1511-1518. doi:10.1038/nn.2684
- Chen, R., Romero, G., Christiansen, M. G., Mohr, A., & Anikeeva, P. (2015). Wireless magnetothermal deep brain stimulation. *Science*, *347*(6229), 1477-1480. doi:10.1126/science.1261821
- Christiansen, M. G., Howe, C. M., Bono, D. C., Perreault, D. J., & Anikeeva, P. (2017). Practical methods for generating alternating magnetic fields for biomedical research. *The Review of scientific instruments*, *88*(8), 084301-084301. doi:10.1063/1.4999358
- Deatsch, A. E., & Evans, B. A. (2014). Heating efficiency in magnetic nanoparticle hyperthermia. *Journal of Magnetism and Magnetic Materials*, *354*, 163-172. doi:https://doi.org/10.1016/j.jmmm.2013.11.006
- Hergt, R., & Dutz, S. (2007). Magnetic particle hyperthermia—biophysical limitations of a visionary tumour therapy. *Journal of Magnetism and Magnetic Materials*, *311*(1), 187-192. doi:https://doi.org/10.1016/j.jmmm.2006.10.1156
- Janssen, M. L., Zwartjes, D. G., Tan, S. K., Vlamings, R., Jahanshahi, A., Heida, T., . . . Temel, Y. (2012). Mild dopaminergic lesions are accompanied by robust changes in subthalamic nucleus activity. *Neurosci Lett*, *508*(2), 101-105. doi:10.1016/j.neulet.2011.12.027

- Johannsmeier, S., Heeger, P., Terakawa, M., Kalies, S., Heisterkamp, A., Ripken, T., & Heinemann, D. (2018). Gold nanoparticle-mediated laser stimulation induces a complex stress response in neuronal cells. *Scientific Reports*, 8(1), 6533. doi:10.1038/s41598-018-24908-9
- Lacroix, L.-M., Carrey, J., & Respaud, M. (2008). A frequency-adjustable electromagnet for hyperthermia measurements on magnetic nanoparticles. *Review of Scientific Instruments*, 79(9), 093909. doi:10.1063/1.2972172
- Meissner, W., Harnack, D., Paul, G., Reum, T., Sohr, R., Morgenstern, R., & Kupsch, A. (2002). Deep brain stimulation of subthalamic neurons increases striatal dopamine metabolism and induces contralateral circling in freely moving 6-hydroxydopamine-lesioned rats. *Neurosci Lett*, 328(2), 105-108. doi:10.1016/s0304-3940(02)00463-9
- Moon, J., Christiansen, M. G., Rao, S., Marcus, C., Bono, D. C., Rosenfeld, D., . . . Anikeeva, P. (2020). Magnetothermal Multiplexing for Selective Remote Control of Cell Signaling. *Advanced Functional Materials*, 30(36), 2000577. doi:https://doi.org/10.1002/adfm.202000577
- Munshi, R., Qadri, S. M., Zhang, Q., Castellanos Rubio, I., del Pino, P., & Pralle, A. (2017). Magnetothermal genetic deep brain stimulation of motor behaviors in awake, freely moving mice. *eLife*, 6, e27069. doi:10.7554/eLife.27069
- Nilius, B., Talavera, K., Owsianik, G., Prenen, J., Droogmans, G., & Voets, T. (2005). Gating of TRP channels: a voltage connection? *The Journal of Physiology*, 567(1), 35-44. doi:https://doi.org/10.1113/jphysiol.2005.088377
- Pankhurst, Q. A., Connolly, J., Jones, S. K., & Dobson, J. (2003). Applications of magnetic nanoparticles in biomedicine. *Journal of Physics D: Applied Physics*, 36(13), R167-R181. doi:10.1088/0022-3727/36/13/201
- Roet, M., Jansen, A., Hoogland, G., Temel, Y., & Jahanshahi, A. (2019). Endogenous TRPV1 expression in the human cingulate- and medial frontal gyrus. *Brain research bulletin*, 152, 184-190. doi:https://doi.org/10.1016/j.brainresbull.2019.07.018
- Rosenfeld, D., Senko, A. W., Moon, J., Yick, I., Varnavides, G., Gregurec, D., . . . Anikeeva, P. (2020). Transgene-free remote magnetothermal regulation of adrenal hormones. *Science Advances*, 6(15), eaaz3734. doi:10.1126/sciadv.aaz3734
- Salvatore, M. F., McInnis, T. R., Cantu, M. A., Apple, D. M., & Pruett, B. S. (2019). Tyrosine Hydroxylase Inhibition in Substantia Nigra Decreases Movement Frequency. *Molecular neurobiology*, 56(4), 2728-2740. doi:10.1007/s12035-018-1256-9
- Suk, J. S., Xu, Q., Kim, N., Hanes, J., & Ensign, L. M. (2016). PEGylation as a strategy for improving nanoparticle-based drug and gene delivery. *Advanced Drug Delivery Reviews*, 99, 28-51. doi:https://doi.org/10.1016/j.addr.2015.09.012
- Voets, T., Droogmans, G., Wissenbach, U., Janssens, A., Flockerzi, V., & Nilius, B. (2004). The principle of temperature-dependent gating in cold- and heat-sensitive TRP channels. *Nature*, 430(7001), 748-754. doi:10.1038/nature02732
- Zhang, H., Shih, J., Zhu, J., & Kotov, N. A. (2012). Layered Nanocomposites from Gold Nanoparticles for Neural Prosthetic Devices. *Nano Letters*, 12(7), 3391-3398. doi:10.1021/nl3015632
- Zhang, Y., Kohler, N., & Zhang, M. (2002). Surface modification of superparamagnetic magnetite nanoparticles and their intracellular uptake. *Biomaterials*, 23(7), 1553-1561. doi:https://doi.org/10.1016/S0142-9612(01)00267-8

Reviewers' Comments:

Reviewer #1:

Remarks to the Author:

The authors answered satisfactorily to my comments.

(I understand that performing ex vivo patch-clamp experiments may indeed be beyond the scope of this study).

Reviewer #2:

Remarks to the Author:

The authors have done a fantastic job at responding to all of my prior questions and comments.

They have also provided new modeling of the depth and breadth of stimulation. Overall, this study provides a novel and important addition to the neuromodulation literature. I would strongly support publication of this paper at this point.

Reviewer #3:

Remarks to the Author:

The authors have addressed my comments, and publication is recommended.